# PolyDiffuse: Polygonal Shape Reconstruction via Guided Set Diffusion Models

**Jiacheng Chen**      **Ruizhi Deng**      **Yasutaka Furukawa**
Simon Fraser University

## Abstract

This paper presents *PolyDiffuse*, a novel structured reconstruction algorithm that transforms visual sensor data into polygonal shapes with Diffusion Models (DM), an emerging machinery amid exploding generative AI, while formulating reconstruction as a generation process conditioned on sensor data. The task of structured reconstruction poses two fundamental challenges to DM: 1) A structured geometry is a "set" (e.g., a set of polygons for a floorplan geometry), where a sample of $N$ elements has $N!$ different but equivalent representations, making the denoising highly ambiguous; and 2) A "reconstruction" task has a single solution, where an initial noise needs to be chosen carefully, while any initial noise works for a generation task. Our technical contribution is the introduction of a Guided Set Diffusion Model where 1) the forward diffusion process learns *guidance networks* to control noise injection so that one representation of a sample remains distinct from its other permutation variants, thus resolving denoising ambiguity; and 2) the reverse denoising process reconstructs polygonal shapes, initialized and directed by the guidance networks, as a conditional generation process subject to the sensor data. We have evaluated our approach for reconstructing two types of polygonal shapes: floorplan as a set of polygons and HD map for autonomous cars as a set of polylines. Through extensive experiments on standard benchmarks, we demonstrate that PolyDiffuse significantly advances the current state of the art and enables broader practical applications. The code and data are available on our project page: `https://poly-diffuse.github.io`.

## 1 Introduction

Reconstruction and generation were once separate research areas with minimal overlaps. While not acknowledged by the broader research communities, state-of-the-art reconstruction and generation techniques have now exhibited increasing similarities. Focusing on structured geometry (e.g., CAD models) in this paper, Auto-regressive Transformer (AR-Transformer) is a family of state-of-the-art generative models that iteratively applies a Transformer network to generate CAD construction sequences [18, 29, 35]. On the reconstruction side, a Transformer network iteratively reconstructs and refines indoor floorplans, outdoor buildings models, and vectorized traffic maps [7, 23, 47], a process resembling the generative AR-Transformer.

In 2023, Generative AI is experiencing significant growth, primarily driven by the emergence of Diffusion Models (DM) [14, 38, 40, 41], where structured geometry generation is not an exception. DM-based approach iteratively denoises coordinates (with associated properties) to generate realistic design layouts [17] or floorplans [36]. A natural question is then "Is a Diffusion Model also good at structured reconstruction?" However, the answer is not that simple.

Take the high-definition (HD) map reconstruction problem for example [22], which aims to reconstruct a set of polylines given sensor data in a bird's-eye view (see Figure 1). A straightforward DM formulation would assume a maximum number of polylines (with a fixed number of corners per poly-

37th Conference on Neural Information Processing Systems (NeurIPS 2023).

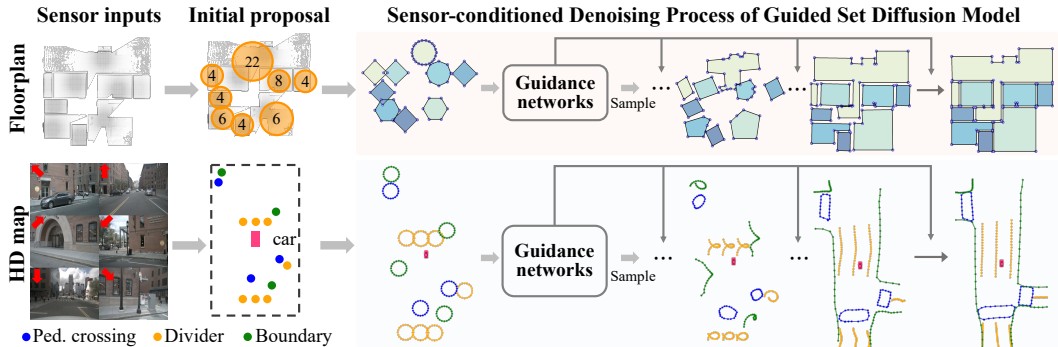

Figure 1: **PolyDiffuse for floorplan and HD map reconstruction.** Starting from an initial proposal (*e.g.*, from a human annotator or an existing method), the sensor-conditioned denoising process of our Guided Set Diffusion Model (GS-DM) generates shape reconstructions in a few sampling steps, initialized and directed by the guidance networks. The initial proposal above mimics simple human inputs that indicate rough locations and specify the number of vertices for the polygonal shapes.

line [23, 25]), take a particular permutation of the elements to construct a fixed-length vector of corner coordinates, then iteratively denoise the vector subject to sensor data as a condition. The structured reconstruction task poses two fundamental challenges to such standard DM formulations: 1) Structured geometry is often a "set" of elements (e.g., a set of polylines), where the geometry of $N$ elements has $N!$ different but equivalent representations, making the denoising factorially ambiguous; 2) The denoising process needs to reach a single solution (*i.e.*, one of $N!$ representations) for a reconstruction task and an initial noise needs to be set carefully, while any initial noise would work for a generation task. Figure 2 shows a toy example of three elements with $6 = 3!$ permutations, where six permutation-equivalent representations of the solution lie in six different sub-spaces. However, they all follow the standard Gaussian and become indistinguishable after the diffusion process, which obfuscates the denoising learning.

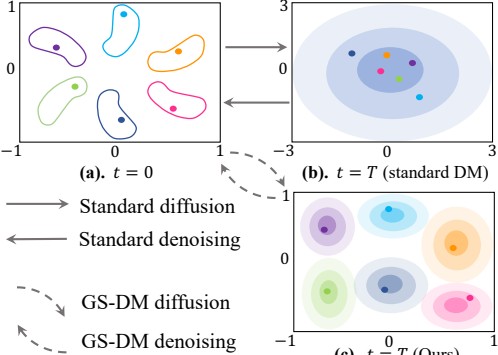

Figure 2: (a) A toy visualization for a sample with three elements, thus 6 different but equivalent points in the data space; (b) Standard DM at $t = T$; (c) Our GS-DM at $t = T$.

Our technical contribution is a Guided Set Diffusion Model (GS-DM). The forward diffusion process learns *guidance networks* controlling noise injection so that one representation of a sample remains separated from its permutation variants, thus resolving denoising ambiguities. Specifically, the guidance networks set an individual Gaussian as the target distribution for each element (instead of a fixed zero-mean unit-variance Gaussian for everything), and are learned before the denoising training by minimizing the permutation ambiguity as a loss function. At test time, the reverse process initializes the element-wise Gaussian noise by the guidance networks and then denoises to reconstruct polygonal shapes conditioned on the sensor data, while the guidance networks take rough initial reconstructions either from an existing method or an annotator (see Figure 1 for examples).

We have conducted extensive experiments on two polygonal structure reconstruction tasks: floorplan reconstruction from a point-cloud density image with Structured3D dataset [48], and HD map construction from onboard camera inputs with nuScenes dataset [2]. PolyDiffuse augments the state-of-the-art approach of the two specific tasks and brings consistent performance improvements, especially on the geometric regularity of the reconstruction results. Our generation process takes 5 to 10 steps to produce decent results and the visual encoder only runs once. We also demonstrate a likelihood-based refinement extension of our system by exploiting the probability flow ODE [41], which enables broader practical applications. To our knowledge, this paper is the first to demonstrate that DM is a powerful machinery for structured reconstruction tasks, in fact, the first in a broader domain of geometry reconstruction as well. We will share all our code and data.

## 2 Background: Denoising Diffusion Probabilistic Models

Diffusion models or score-based generative models [14, 38, 40, 41] progressively inject noise to data in the forward (diffusion) process and generate data from noise by the reverse (denoising) process. The section provides the key equations on the denoising diffusion probabilistic models (DDPM) [14] that lay the foundation of our method.

DDPM considers a Markovian forward process that turns data $\mathbf{x}_0 \sim q(\mathbf{x}_0)$ into Gaussian noise:

$$q(\mathbf{x}_t|\mathbf{x}_{t-1}) := \mathcal{N}(\mathbf{x}_t; \ \sqrt{1-\beta_t}\mathbf{x}_{t-1}, \ \beta_t\mathbf{I}) \tag{1}$$

$t = 1, \ldots, T$ and $\mathbf{x}_t$ is the latent at $t$. The noise schedule $\{\beta_t\}$ is constant [14, 28] or learned by reparameterization [20]. The above forward process is written in closed form for an arbitrary $t$:

$$q(\mathbf{x}_t|\mathbf{x}_0) = \mathcal{N}(\mathbf{x}_t; \sqrt{\bar{\alpha}_t}\mathbf{x}_0, (1-\bar{\alpha}_t)\mathbf{I}), \tag{2}$$

$$\bar{\alpha}_0 = 1, \quad \bar{\alpha}_t := \bar{\alpha}_{t-1}\alpha_t, \quad \alpha_t := 1 - \beta_t. \tag{3}$$

$\mathbf{x}_t$ is sampled by $\mathbf{x}_t = \sqrt{\bar{\alpha}_t}\mathbf{x}_0 + \sqrt{(1-\bar{\alpha}_t)}\boldsymbol{\epsilon}$ for $\boldsymbol{\epsilon} \sim \mathcal{N}(\mathbf{0}, \mathbf{I})$. DDPM parameterizes the reverse process with a noise prediction (or denoising) network $\boldsymbol{\epsilon}_\theta(\mathbf{x}_t, t)$ to make connections with denoising score matching and Langevin dynamics [40, 45], and the sampling step of the reverse process is derived as:

$$\mathbf{x}_{t-1} = \frac{1}{\sqrt{\alpha_t}}\left[\mathbf{x}_t - \frac{1-\alpha_t}{\sqrt{1-\bar{\alpha}_t}}\boldsymbol{\epsilon}_\theta(\mathbf{x}_t, t)\right] + \sigma_t\mathbf{z}, \ \mathbf{z} \sim \mathcal{N}(\mathbf{0}, \mathbf{I}). \tag{4}$$

$\sigma_t^2$ is set to $\beta_t$ or $\frac{1-\bar{\alpha}_{t-1}}{1-\bar{\alpha}_t}\beta_t$. The final training objective is a reweighted variational lower bound:

$$L_{\text{simple}}(\theta) := \mathbb{E}_{\mathbf{x}_0, t, \boldsymbol{\epsilon}}\left[||\boldsymbol{\epsilon} - \boldsymbol{\epsilon}_\theta(\sqrt{\bar{\alpha}_t}\mathbf{x}_0 + \sqrt{1-\bar{\alpha}_t}\boldsymbol{\epsilon}, t)||^2\right] \tag{5}$$

## 3 Guided Set Diffusion Models (GS-DM)

Consider the task of generating a set of elements $\mathbf{x} = \{\mathbf{x}^1, \ldots, \mathbf{x}^N\}$ under a condition $\mathbf{y}$. A straightforward application of diffusion models would be to take one permutation of the elements to form a vector, then perform diffusion and learn denoising. With an abuse of notation, we let the vector $\mathbf{x}_0$ (*i.e.*, the sample or denoised result in DM's notation) represent one particular permutation of $\mathbf{x}$. The challenge is that a set of $N$ elements has $N!$ different but equivalent representations, and any one of the $N!$ is a valid $\mathbf{x}_0$, making the denoising factorially ambiguous [1]. To alleviate the above issue, our idea learns to guide the noise injection in a per-element manner by learning two "guidance networks", such that a sample $\mathbf{x}_0$ remains separated from its permutation variants throughout the diffusion process. At test time, the guidance networks produce per-element Gaussians and stepwise guidance from an initial reconstruction, from which the denoising process generates the final reconstruction iteratively conditioned on sensor data. We first explain the forward and reverse processes and then the two-stage training for the guidance and denoising networks.

**Forward process**: With the standard diffusion process, different permutation variants of a sample gradually become indistinguishable through the diffusion process. To keep a particular representation $\mathbf{x}_0$ separated from its permutation variants, we propose to inject noise per element. Let $\mathbf{x}_t^i$ denote the diffused element $\mathbf{x}_0^i$ at timestep $t$, we learn to change the transition distribution $q(\mathbf{x}_t^i|\mathbf{x}_{t-1}^i)$ by adding $\boldsymbol{\mu}_\phi(\mathbf{x}_0, t, i)$ to the mean and multiplying $\boldsymbol{\sigma}_\phi(\mathbf{x}_0, t, i)$ to the standard deviation:

$$q(\mathbf{x}_t^i|\mathbf{x}_{t-1}^i, \mathbf{x}_0) := \mathcal{N}(\mathbf{x}_t^i; \ \sqrt{1-\beta_t}\mathbf{x}_{t-1}^i + \boldsymbol{\mu}_\phi(\mathbf{x}_0, t, i), \ \beta_t\boldsymbol{\sigma}_\phi^2(\mathbf{x}_0, t, i)\mathbf{I}) \tag{6}$$

$\boldsymbol{\mu}_\phi$ and $\boldsymbol{\sigma}_\phi$ take $\mathbf{x}_0$ and produce outputs for i[th] element at $t$. Following the similar derivations as in DDPM [14], we obtain (see Appendix A.1 for derivations):

$$q(\mathbf{x}_t^i|\mathbf{x}_0^i) = \mathcal{N}(\mathbf{x}_t^i; \ \sqrt{\bar{\alpha}_t}\mathbf{x}_0^i + \bar{\boldsymbol{\mu}}_\phi(\mathbf{x}_0, t, i), \ \bar{\boldsymbol{\sigma}}_\phi^2(\mathbf{x}_0, t, i)\mathbf{I}), \tag{7}$$

$$\bar{\boldsymbol{\mu}}_\phi(\mathbf{x}_0, 0, i) = 0, \quad \bar{\boldsymbol{\mu}}_\phi(\mathbf{x}_0, t, i) := \sqrt{\alpha_t}\bar{\boldsymbol{\mu}}_\phi(\mathbf{x}_0, t-1, i) + \boldsymbol{\mu}_\phi(\mathbf{x}_0, t, i), \tag{8}$$

$$\bar{\boldsymbol{\sigma}}_\phi^2(\mathbf{x}_0, 0, i) = 0, \quad \bar{\boldsymbol{\sigma}}_\phi^2(\mathbf{x}_0, t, i) := \alpha_t\bar{\boldsymbol{\sigma}}_\phi^2(\mathbf{x}_0, t-1, i) + (1-\alpha_t)\boldsymbol{\sigma}_\phi^2(\mathbf{x}_0, t, i). \tag{9}$$

---

[1]This permutation ambiguity issue persists with permutation-invariant models (*e.g.*, Transformers), as it is inevitable to pick one particular permutation of $\mathbf{x}$ as the concrete data representation $\mathbf{x}_0$.

| **Algorithm 1** Guidance training (stage 1) | **Algorithm 2** Denoising training (stage 2) |
|---|---|
| 1: $\phi$: trainable, $\theta$: frozen | 1: $\phi$: frozen, $\theta$: trainable |
| 2: **repeat** | 2: **repeat** |
| 3: $\quad \mathbf{x}_0 \sim q(\mathbf{x}_0)$ | 3: $\quad \mathbf{x}_0 \sim q(\mathbf{x}_0)$ |
| 4: $\quad t \sim \text{Uniform}(\{1, \dots, T\})$ | 4: $\quad t \sim \text{Uniform}(\{1, \dots, T\})$ |
| 5: $\quad \boldsymbol{\epsilon} \sim \mathcal{N}(\mathbf{0}, \mathbf{I})$, then compute $\mathbf{x}_t$ | 5: $\quad \boldsymbol{\epsilon} \sim \mathcal{N}(\mathbf{0}, \mathbf{I})$, then compute $\mathbf{x}_t$ |
| 6: $\quad$ Take gradient descent step on $\nabla_\phi L_{\text{guide}}$ (Eq.13) | 6: $\quad$ Take gradient descent step on $\nabla_\theta L_{\text{simple}}$ (Eq.11) |
| 7: **until** converged | 7: **until** converged |

As $q(\mathbf{x}_t^i | \mathbf{x}_0^i)$ only explicitly depends on $\bar{\boldsymbol{\mu}}_\phi$ and $\bar{\boldsymbol{\sigma}}_\phi$, and $\mathbf{x}_T^i \sim \mathcal{N}(\bar{\boldsymbol{\mu}}_\phi(\mathbf{x}_0, T, i), \bar{\boldsymbol{\sigma}}_\phi^2(\mathbf{x}_0, T, i))$, we consider $\bar{\boldsymbol{\mu}}_\phi$ and $\bar{\boldsymbol{\sigma}}_\phi$ as the guidance networks. $\boldsymbol{\mu}_\phi$ and $\boldsymbol{\sigma}_\phi$ are defined implicitly by Eq. 8 and Eq. 9. To simplify the formulation, we define $\bar{\sigma}_\phi(\mathbf{x}_0, t, i) := \sqrt{1 - \bar{\alpha}_t} C(\mathbf{x}_0, i)$ where $C(\mathbf{x}_0, i)$ is a function independent of the timestep $t$, thus implicitly defining $\boldsymbol{\sigma}_\phi(\mathbf{x}_0, t, i) = \bar{\boldsymbol{\sigma}}_\phi(\mathbf{x}_0, T, i) = C(\mathbf{x}_0, i)$.

**Reverse process**: Note that $\mathbf{x}_0$ is the ground truth and only available for training. At test time, we introduce a *proposal generator* to produce an initial reconstruction $\hat{\mathbf{x}}_0$, and run $\bar{\boldsymbol{\mu}}_\phi$ and $\bar{\boldsymbol{\sigma}}_\phi$ with $\hat{\mathbf{x}}_0$. $\hat{\mathbf{x}}_0$ can either be results from an existing method or rough annotations from a human annotator. We first draw the element-wise initial noise with $\mathbf{x}_T^i \sim \mathcal{N}(\bar{\boldsymbol{\mu}}_\phi(\hat{\mathbf{x}}_0, T, i), \bar{\boldsymbol{\sigma}}_\phi^2(\hat{\mathbf{x}}_0, T, i))$, and then run denoising steps iteratively. Similar to Eq.4, the sampling step of the reverse process is derived as

$$\mathbf{x}_{t-1}^i = \frac{1}{\sqrt{\alpha_t}} \left[ \mathbf{x}_t^i - \bar{\boldsymbol{\mu}}_\phi(\hat{\mathbf{x}}_0, t, i) - \bar{\boldsymbol{\sigma}}_\phi(\hat{\mathbf{x}}_0, t, i) \frac{1 - \alpha_t}{1 - \bar{\alpha}_t} \boldsymbol{\epsilon}_\theta^i(\mathbf{x}_t, t, \mathbf{y}) \right] + \bar{\boldsymbol{\mu}}_\phi(\hat{\mathbf{x}}_0, t - 1, i) + \boldsymbol{\sigma}_t \mathbf{z}^i \quad (10)$$

The denoising network $\boldsymbol{\epsilon}_\theta$ takes the entire $\mathbf{x}_t$ (*i.e.*, all elements), and $\boldsymbol{\epsilon}_\theta^i$ is the output of the i$^{\text{th}}$ element. Please see Appendix A.2 for derivations.

**Learning the denoising network**: Following Eq. 5 of standard DDPM, the denoising objective is

$$L_{\text{simple}}(\theta) := \mathbb{E}_{\mathbf{x}_0, t, \boldsymbol{\epsilon}} \left[ \sum_{i=1}^N ||\boldsymbol{\epsilon}^i - \boldsymbol{\epsilon}_\theta^i(\{\sqrt{\bar{\alpha}_t} \mathbf{x}_0^i + \bar{\boldsymbol{\mu}}_\phi(\mathbf{x}_0, i, t) + \bar{\boldsymbol{\sigma}}_\phi(\mathbf{x}_0, i, t) \boldsymbol{\epsilon}^i\}, t, \mathbf{y})||^2 \right] \quad (11)$$

The above formulation from Eq.6 to Eq.11 resembles that of DDPM in §2, where the key difference is the introduction of guidance networks $\bar{\boldsymbol{\mu}}_\phi$ and $\bar{\boldsymbol{\sigma}}_\phi$. We then describe the training of the guidance networks, which is the key step of GS-DM.

**Learning the guidance network**: Before training the denoising network with Eq. 11, we train $\bar{\boldsymbol{\mu}}_\phi$ and $\bar{\boldsymbol{\sigma}}_\phi$ to ensure that $\mathbf{x}_t$ in the diffusion process keep separated from other permutation variants of $\mathbf{x}_0$. We quantify the permutation invariance between $\mathbf{x}_0$ and $\mathbf{x}_t$ with a triplet loss $L_{\text{Triplet}}(\cdot, \cdot, \cdot)$ widely used in metric learning [4, 34] (full implementation details are in Appendix B.3):

$$L_{\text{perm}}(\phi) := \max_{\mathbf{x}_0' \in \mathcal{P}^!(\mathbf{x}_0) \setminus \{\mathbf{x}_0\}} L_{\text{Triplet}}(\mathbf{x}_t, \mathbf{x}_0, \mathbf{x}_0') + \max_{\mathbf{x}_t' \in \mathcal{P}^!(\mathbf{x}_t) \setminus \{\mathbf{x}_t\}} L_{\text{Triplet}}(\mathbf{x}_0, \mathbf{x}_t, \mathbf{x}_t') \quad (12)$$

$\mathcal{P}^!(\mathbf{x}_0)$ is the set of all permutation variants of $\mathbf{x}_0$. Using the terminology in metric learning, the three inputs of $L_{\text{Triplet}}(\cdot, \cdot, \cdot)$ are the anchor, positive, and negative, respectively. Our loss only considers the closest negative permutation, which is similar to the hard negative mining in some variants of the triplet loss [6, 10]. We also add two regularization terms on $\bar{\boldsymbol{\mu}}_\phi$ and $\bar{\boldsymbol{\sigma}}_\phi$, so that the means of all elements are not too scattered and the variances do not vanish. The final loss for training the guidance networks is:

$$L_{\text{guide}}(\phi) := \mathbb{E}_{t, \mathbf{x}_0, \boldsymbol{\epsilon}} \left[ \lambda_1 L_{\text{perm}}(\phi) + \lambda_2 \sum_{i=1}^N ||1/\bar{\boldsymbol{\sigma}}_\phi(\mathbf{x}_0, t, i)||^2 + \lambda_3 \sum_{i=1}^N ||\bar{\boldsymbol{\mu}}_\phi(\mathbf{x}_0, t, i)||^2 \right] \quad (13)$$

Algorithm 1 and Algorithm 2 summarize the two-stage training paradigm of GS-DM.

## 4 PolyDiffuse: Polygonal shape reconstruction via GS-DM

PolyDiffuse uses the GS-DM to solve polygonal shape reconstruction tasks (see Figure 3), in particular, floorplan and HD map reconstruction. This section explains specific details and specializations.

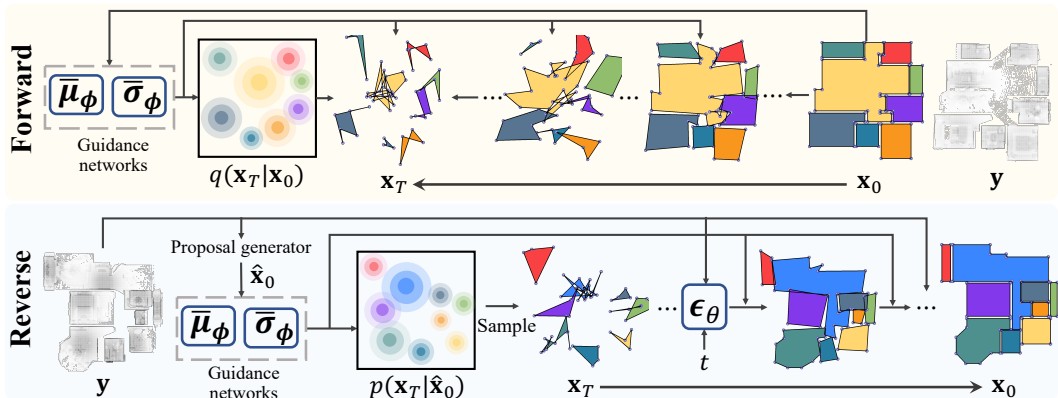

Figure 3: Illustration of the forward and reverse processes of PolyDiffuse with floorplan data.

**Feature representation**: A sample $\mathbf{x}$ is a set of elements $\mathbf{x}^i$ as defined in §3. Each element $\mathbf{x}^i$ is either a (closed) polygon or a polyline, consisting of an arbitrary (for the floorplan task) or a fixed (for the HD map task) number of vertices as a sequence: $\mathbf{x}^i = \left[v_1^i, \ldots, v_{N_i}^i\right]$. Each vertex contains a 2D coordinate. For the ground-truth data, we determine the first vertex of each element by sorting based on the Y-axis coordinate and then the X-axis coordinate, similarly to PolyGen [27]. The polygons are always in a counter-clockwise orientation.

**Guidance network**: As derived in §3, the forward (Eq.7) and reverse (Eq.10) processes only depend on $\bar{\boldsymbol{\mu}}_\phi$ and $\bar{\boldsymbol{\sigma}}_\phi$, and each element follows $\mathbf{x}_T^i \sim \mathcal{N}(\bar{\boldsymbol{\mu}}_\phi(\mathbf{x}_0, T, i), \bar{\boldsymbol{\sigma}}_\phi^2(\mathbf{x}_0, T, i))$. Therefore, we choose to directly parameterize $\bar{\boldsymbol{\mu}}_\phi(\mathbf{x}_0, T, i) = \text{Trans}_{\bar{\boldsymbol{\mu}}_\phi}(\mathbf{x}_0, i)$ and $\bar{\boldsymbol{\sigma}}_\phi(\mathbf{x}_0, T, i) = \text{Trans}_{\bar{\boldsymbol{\sigma}}_\phi}(\mathbf{x}_0, i)$ as two Transformer [44] networks, and define for all timesteps $t = 1, \ldots, T-1$ as follows:

$$\bar{\boldsymbol{\mu}}_\phi(\mathbf{x}_0, t, i) := (1 - \sqrt{\bar{\alpha}_t}) \text{Trans}_{\bar{\boldsymbol{\mu}}_\phi}(\mathbf{x}_0, i), \quad \bar{\boldsymbol{\sigma}}_\phi(\mathbf{x}_0, t, i) := \sqrt{1 - \bar{\alpha}_t} \text{Trans}_{\bar{\boldsymbol{\sigma}}_\phi}(\mathbf{x}_0, i). \quad (14)$$

Note that Eq.14 makes the noise adaptation directly dependent on $\bar{\alpha}_t$ for simplicity and drops $t$ from the input. Algorithm 1 thus learns the above two Transformers. As for the proposal generator to produce $\hat{\mathbf{x}}_0$ at test time, we either employ the state-of-the-art task-specific method (*i.e.*, Room-Former [47] for floorplan and MapTR [23] for HD map) or simulate rough human annotations. Please see §5 for experiments with different proposal generators, and Appendix B.1 for implementation details of the guidance network.

**Denoising network**: The implementation of the denoising network $\boldsymbol{\epsilon}_\theta(\mathbf{x}_t, t, \mathbf{y})$ borrows the core neural architectures from the state-of-the-art task-specific models RoomFormer [47] and MapTR [23]. While these two models are both adapted from DETR [3] and learn direct mappings from $\mathbf{y}$ to $\mathbf{x}$ with encoder-decoder Transformers, we need to make a few essential modifications to turn them into valid denoising networks (full implementation details are in Appendix B.2):

- Instead of the learnable embeddings or coordinates, $\mathbf{x}_t$ should be the input nodes to the Transformer decoder. Each vertex is a node, and the node feature is the concatenation of four positional encodings [44]: two for X and Y-axis coordinates and another two for vertex and instance index.
- We encode the timestep $t$ (or noise level) with a positional encoding [9, 14], and add this feature to each block of the Transformer decoder.

Our overall formulation with GS-DM is independent of the network architectures, and more advanced networks can be easily integrated into the framework in the future to boost performance.

**Sampling acceleration**: PolyDiffuse follows Eq.10 to reconstruct polygonal shapes. Song et al.[41] shows that DDPM can be viewed as a discretization of a stochastic differential equation (SDE) with an associated probability flow ODE, and advanced ODE solvers [19, 26, 39] can significantly accelerate the sampling process. At test time, we employ a first-order solver (*e.g.*, DDIM [39]) and use 10 sampling steps for both tasks (ablation study is in §5.3). During the generation process, PolyDiffuse only runs the image encoder once as the visual features are shared across all steps.

**Likelihood evaluation**: With the probability flow ODE [41], PolyDiffuse can estimate its own reconstruction quality via likelihood evaluation, thus enabling potential extensions such as search-based or human-in-the-loop refinement as a wrapper around our system. This is an exciting property not possessed by previous reconstruction approaches. We show examples in Figure 4.

# 5 Experiments

We have implemented the system with PyTorch and used a machine with 4 NVIDIA RTX A5000 GPUs. We have borrowed the official codebase of Karras et al.[19] to implement the diffusion model framework. The guidance networks are implemented as set-to-set Transformer decoders, and the loss weights for the guidance training are $\lambda_1 = 1, \lambda_2 = 0.05, \lambda_3 = 0.1$. The implementation of the denoising network refers to the competing state-of-the-art, RoomFormer [47] and MapTR [23] for floorplan and HD map reconstruction, respectively. Concretely, we employ the same ResNet [12] image encoder, DETR-style Transformer architectures (discussed in §4), learning rate settings, and optimizer settings as RoomFormer or MapTR, while increasing the number of training iterations ($\times 2.5$ for floorplan and $\times 1.5$ for HD map), as the model converges slower under a denoising formulation than a detection/regression formulation. Note that we tried the same training schedule for the competing methods but their performance dropped due to overfitting (see §5.3 for ablation study). For each task, we use the same trained model for different types of proposals (*i.e.*, existing methods or rough annotations). Complete implementation details are in Appendix B, and supplementary experiments are in Appendix D

## 5.1 Floorplan reconstruction

**Dataset and metrics**: Structured3D dataset [48] contains 3500 indoor scenes (3000/250/250 for training/validation/test) with diverse house floorplans. The average/maximum numbers of polygons and vertex per polygon across the dataset are 6.29/17 and 5.72/38, respectively. Point clouds are converted to $256 \times 256$ top-view point-density images as inputs. We use the same evaluation metrics as previous works [7, 42, 47], which consists of three levels of metrics with increasing difficulty: room, corner, and angle. The precision, recall, and F1 score are reported at each metric level.

**Competing approaches**: We compare PolyDiffuse with five approaches from the literature: Floor-SP [5], MonteFloor [42], LETR [46], HEAT [7], and RoomFormer [47]. The first two approaches design sophisticated optimization algorithms to solve for the optimal floorplan with learned metric functions, while the latter three use end-to-end Transformer-based neural networks, thus being much faster. For our proposal generator RoomFormer, we simplify the implementation by removing the Dice loss. As the dataset is very small, we add a random rotation data augmentation and train the modified RoomFormer for $4\times$ the original training iterations, which improves the overall performance.

Table 1: **Quantitative evaluation on Structured3D test set [48].** PolyDiffuse outperforms the state-of-the-art methods by clear margins, and works well with rough annotations. Results of previous works are copied from [47]. $^*$ indicates our modified implementation. $^\dagger$: The running time is measured on a single Nvidia RTX A5000 GPU, and we only report those run by ourselves. When using RoomFormer to produce the initial proposals, the running time counts both the RoomFormer proposal generator and the GS-DM.

| Evaluation Level → | | | | Room | | | Corner | | | Angle | | |
|---|---|---|---|---|---|---|---|---|---|---|---|---|
| Method | Stages | Steps | FPS$^\dagger$ | Prec. | Rec. | F1 | Prec. | Rec. | F1 | Prec. | Rec. | F1 |
| Floor-SP [5] | 2 | - | - | 89. | 88. | 88. | 81. | 73. | 76. | 80. | 72. | 75. |
| MonteFloor [42] | 2 | 500 | - | 95.6 | 94.4 | 95.0 | 88.5 | 77.2 | 82.5 | 86.3 | 75.4 | 80.5 |
| LETR [46] | 1 | 1 | - | 94.5 | 90.0 | 92.2 | 79.7 | 78.2 | 78.9 | 72.5 | 71.3 | 71.9 |
| HEAT [7] | 2 | 3 | - | 96.9 | 94.0 | 95.4 | 81.7 | 83.2 | 82.5 | 77.6 | 79.0 | 78.3 |
| RoomFormer [47] | 1 | 1 | - | 97.9 | 96.7 | 97.3 | 89.1 | 85.3 | 87.2 | 83.0 | 79.5 | 81.2 |
| RoomFormer$^*$ | 1 | 1 | 29.9 | 96.3 | 96.2 | 96.2 | 89.7 | 86.7 | 88.2 | 85.4 | 82.5 | 83.9 |
| +PolyDiffuse(Ours) | 2 | 2 | 11.7 | 96.9 | 96.4 | 96.6 | 90.3 | 87.1 | 88.7 | 85.8 | 82.8 | 84.3 |
| +PolyDiffuse(Ours) | 2 | 5 | 7.1 | 98.5 | 97.9 | 98.2 | 92.5 | 89.0 | 90.7 | 90.3 | 86.9 | 88.6 |
| +PolyDiffuse(Ours) | 2 | 10 | 4.4 | **98.7** | **98.1** | **98.4** | **92.8** | **89.3** | **91.0** | **90.8** | **87.4** | **89.1** |
| Rough annotations | 1 | - | - | 17.9 | 18.2 | 18.0 | 1.3 | 1.4 | 1.3 | 0.1 | 0.1 | 0.1 |
| +PolyDiffuse(Ours) | 2 | 10 | 19.2 | 97.4 | 98.2 | 97.8 | 91.7 | 92.2 | 91.9 | 89.2 | 89.7 | 89.4 |

**Quantitative & qualitative evaluation**: Table 1 presents the quantitative evaluation results. We tried two proposal generators: 1) the improved RoomFormer and 2) rough annotations. The rough annotations are prepared by turning the ground-truth data into fixed-radius small circles centered

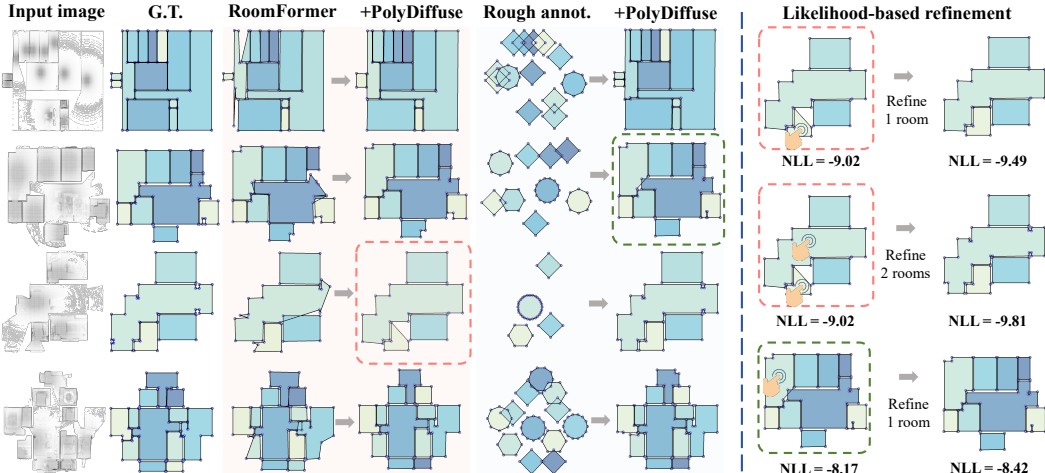

Figure 4: **Qualitative results for floorplan reconstruction.** *Left:* PolyDiffuse improves the geometry of RoomFormer and turns rough annotations into decent reconstructions. *Right:* PolyDiffuse enables search-based self-refinement by enumerating different vertex numbers of polygons and evaluating its own reconstruction likelihood (*i.e.*, NLL per dimension).

at the instance centroid (see Figure 4), which simulates what a human annotator can provide easily (*i.e.*, indicating instance center and the number of vertices by looking at images). When using RoomFormer as the proposal generator, PolyDiffuse outperforms all competing methods by clear margins across all the evaluation metrics. The performance gap increases with the difficulty level of the metric, indicating that PolyDiffuse is superior in high-level geometry reasoning. With the rough annotations as the initial proposal, PolyDiffuse also produces reasonable reconstructions. The recalls are especially high because of the correct number of vertices. Note that our training is not dependent on any specific proposal generator, but PolyDiffuse works well with different generators at test time. Figure 4 presents concrete qualitative examples of how PolyDiffuse improves the initial reconstruction and enforces better geometry correctness. We also provide more qualitative examples in Appendix D.3.

**Running speed analysis**: Table 1 also presents the speed-performance tradeoff of PolyDiffuse on the floorplan reconstruction task. During the denoising process, the image encoding parts of the denoising network only run once, while the transformer decoder part runs for multiple rounds. In our computation environment, the time of image encoding vs. the time of transformer decoder is 2:1. Since speed is not a crucial factor for floorplan reconstruction, we can just use more denoising steps in real applications for better reconstruction quality.

**Likelihood-based refinement**: Figure 4(right) demonstrates how we leverage the likelihood evaluation to refine reconstruction results with minimal user interaction. We manually indicate the imperfect polygons (inside dashed boxes at the left), then run our system with different numbers of vertices for these polygons while evaluating the reconstruction likelihood. The result with the best negative log-likelihood (NLL) per dimension is the refined reconstruction.

## 5.2 HD map reconstruction

**Dataset and metrics**: The nuScenes dataset [2] provides a standard benchmark for HD map reconstruction. The dataset contains 1000 ego-centric sequences collected by autonomous vehicles with rich annotations. The data is annotated at 2Hz. Each sample has 6 RGB images and LiDAR sweeps captured by onboard sensors, covering the $360°$ horizontal FOV. In the top-down Bird-eye-view (BEV) space, the perception range is $[-15m, 15m]$ for X-axis and $[-30m, 30m]$ for Y-axis. The map includes three categories of elements: pedestrian crossing, divider, and road boundary. For fair comparisons, we use the same dataset split and pre-processing setups as previous works [22, 23, 25], and also represent each polyline with 20 uniformly-interpolated vertices. Similar to MapTR [23], we only use the RGB images as the sensor inputs in our experiments.

We follow the common evaluation protocol from the literature [23, 25], which employs the average precision (AP) with the Chamfer distance as a matching criterion. The AP is averaged over three

Table 2: **Quantitative evaluation reusls of HD map construction on nuScenes[2].** All methods in the table use RGB inputs only and employ ResNet50 as the image backbone. [+]: Results are obtained by running the official code with the released model checkpoint. [†]: The running time here is measured on a single Nvidia RTX A5000 GPU, and we only report for the methods run by ourselves. When using MapTR to produce the proposals, the running time counts both the MapTR proposal generator and the GS-DM.

| Matching Criterion → | | | | Chamfer distance | | | | + Ordered angle distance | | | |
| --- | --- | --- | --- | --- | --- | --- | --- | --- | --- | --- | --- |
| Method | Stages | Steps | FPS[†] | $AP_p$ | $AP_d$ | $AP_b$ | mAP | $AP_p$ | $AP_d$ | $AP_b$ | mAP |
| VectorMapNet [25] | 1 | 20 | - | 36.1 | 47.3 | 39.3 | 40.9 | - | - | - | - |
| VectorMapNet [25] | 2 | 20 | - | 42.5 | 51.4 | 44.1 | 46.0 | - | - | - | - |
| VectorMapNet[+] | 1 | 20 | 3.9 | 40.0 | 47.6 | 39.0 | 42.2 | 33.0 | 44.5 | 27.3 | 34.9 |
| MapTR [23] | 1 | 1 | - | 56.2 | 59.8 | 60.1 | 58.7 | - | - | - | - |
| MapTR[+] | 1 | 1 | 14.3 | 55.8 | **60.9** | 61.1 | 59.3 | 46.1 | 43.4 | 41.9 | 43.8 |
| +PolyDiffuse(Ours) | 2 | 2 | 6.3 | 56.8 | 59.8 | 60.9 | 59.2 | 50.3 | 48.2 | 44.3 | 47.6 |
| +PolyDiffuse(Ours) | 2 | 5 | 4.8 | 58.1 | 59.7 | 61.2 | 59.6 | 51.8 | 49.5 | 45.4 | 48.9 |
| +PolyDiffuse(Ours) | 2 | 10 | 3.4 | **58.2** | 59.7 | **61.3** | **59.7** | **52.0** | **49.5** | **45.4** | **49.0** |
| Rough annotations | - | - | - | 18.5 | 2.2 | 0.7 | 7.1 | 8.7 | 0.0 | 0.0 | 2.9 |
| +PolyDiffuse(Ours) | 2 | 10 | 11.3 | 55.3 | 60.4 | 55.3 | 57.0 | 48.3 | 49.5 | 38.1 | 45.3 |

distance thresholds: $\{0.5m, 1.0m, 1.5m\}$, and the mean average precision (mAP) is obtained by averaging across the three classes of map elements. However, the Chamfer distance does not consider the vertex order and vertex matching between prediction and G.T., thus failing to measure the structured correctness. As directional information is critical for autonomous vehicles, we propose to augment the matching criterion with an extra order-aware angle distance. Concretely, we find the optimal vertex matching between a prediction and a G.T. with the smallest coordinate distance, then trace along the two polylines/polygons to compute the average angle distance (in degrees). A prediction is considered a true positive only when satisfying both the Chamfer and angle distance thresholds. We augment the three-level thresholds to $\{(0.5m, 5°), (1.0m, 10°), (1.5m, 15°)\}$. Results are reported in both the original and augmented metrics. We provide complete implementation details and visualizations about the augmented metric in the Appendix C.

**Competing approaches**: We compare our method with VectorMapNet [25] and MapTR [23]. VectorMapNet [25] employs an auto-regressive transformer decoder to generate the vertex of polygon/polyline one by one, conditioned on the image inputs. MapTR [23] is our proposal generator and is a DETR-style hierarchical detection method that directly estimates the polygon/polyline as a set of vertex sequences. MapTR and RoomFormer are technically very similar but focus on different tasks.

**Quantitative & qualitative evaluation**: Table 2 presents the quantitative evaluation results. PolyDiffuse does not significantly improve the Chamfer-distance mAP of MapTR, as it does not discover new instances. However, it shows clear advantages in the order-aware angle distance, a metric that is indicative of high-level structural regularities. Figure 5 qualitatively shows that PolyDiffuse accurately reconstructs curves with greater resemblance to the ground truth. We also experimented with the same rough annotations as in the floorplan task, and the quantitative results show that the mAP becomes lower compared to using MapTR as the proposal generator. By analyzing the results with rough annotations more carefully, we found the AP with the lowest threshold becomes much worse, while the one with the highest threshold improves. We believe this is because the guidance networks are trained with "G.T. proposals", where the rough annotations (*i.e.*, fixed radius circles) have a very different curve style when serving as the proposals, thus producing inaccurate initial Gaussians and stepwise guidance. This is a potential limitation of the current method. We provide additional qualitative examples in Appendix D.3.

**Running speed analysis**: Table 2 also presents our speed-performance tradeoff on the HD map reconstruction task. In our computation environment, the time of image encoding vs. the time of transformer decoder is 4:1 when using MapTR's model architecture. As FPS is an important consideration for online applications, we need to pick the number of denoising steps carefully. Furthermore, since PolyDiffuse is not restricted to a specific task-specific method for the model

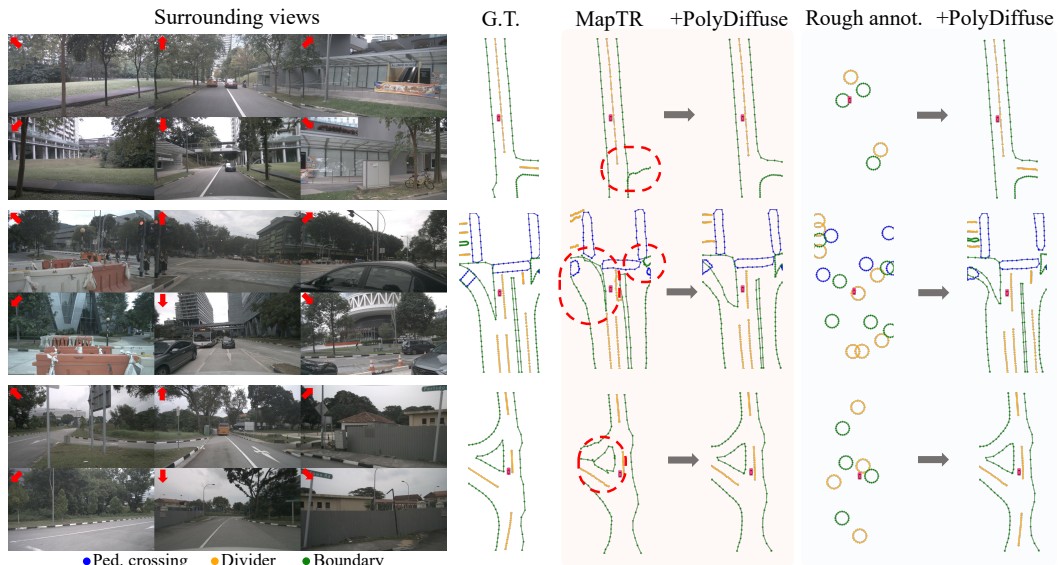

| Surrounding views | G.T. | MapTR | +PolyDiffuse | Rough annot. | +PolyDiffuse |

●Ped. crossing  ●Divider  ●Boundary

Figure 5: **Qualitative results for HD map reconstruction.** PolyDiffuse improves the geometry correctness of MapTR (marked in red) and can turn rough annotations into reasonable reconstructions.

architecture and proposal generator, our performance/efficiency can keep improving when better task-specific base methods appear in the future.

### 5.3 Ablation studies

We provide key ablation studies with the floorplan reconstruction benchmark. Please also find additional ablation studies in Appendix D.1.

**Standard DM vs. GS-DM**: To better understand the effectiveness of our GS-DM, we define a standard DM using a vanilla DDPM with images as the condition. The standard DM does not have the guidance network and proposal generator. The reverse process starts with a noise sampled from the standard Gaussian distribution and gradually denoises it into the final reconstruction using the same sampler as GS-DM. It also has the same denoising network architecture as the GS-DM of PolyDiffuse. As shown in Table 3, although standard DM produces reasonable results with 10 sampling steps, it does not outperform the baseline method RoomFormer. Also, the performance of standard DM drops significantly with fewer sampling steps, as the denoising ambiguity hurts the performance more seriously with a larger sampling step size. We also demonstrate in Appendix D.2 that the standard DM can fail in a toy experiment due to denoising ambiguity.

**Sampling steps**: Unlike the standard DM, Table 1, Table 2, and Table 3 all suggest that PolyDiffuse is robust to fewer sampling steps and can still slightly improve RoomFormer or MapTR with a very small number of sampling steps.

**Training epochs**: We observed that the model converges slower under the denoising formulation than the detection/regression formulation of previous approaches. In Table 4, PolyDiffuse keeps improving when training with longer schedules, while RoomFormer suffers from overfitting. However, even with the default schedule, PolyDiffuse significantly outperforms the baseline.

## 6 Related Work

**Structured reconstruction of polygonal shapes**: Structured geometry reconstruction is an active research area in computer vision, focusing on converting raster sensor data into vectorized geometries such as room layouts [8, 13], floorplans [1, 24], planes [11, 37], wireframes [16], *etc*. Taking advantage of the advancements in deep learning, methods with end-to-end neural architecture [7, 23, 25, 47, 49, 51] gradually dominate the area with superior accuracy and efficiency. In this paper, we formulate the reconstruction of polygonal shapes as a conditioned generation process with diffusion models, one of the essential machinery of the recent exploding generative AI. Our

Table 3: Comparison between standard DM and GS-DM with different sampling steps. Results are reported in F1 scores.

| Method | Steps | Room | Corner | Edge |
|---|---|---|---|---|
| RoomFormer | 1 | 96.2 | 88.2 | 83.9 |
| | 2 | 12.5 | 6.5 | 5.4 |
| Standard DM | 5 | 86.7 | 79.6 | 77.0 |
| | 10 | 90.4 | 83.6 | 81.1 |
| PolyDiffuse (w/ GS-DM) | 2 | 96.6 | 88.7 | 84.3 |
| | 5 | 98.1 | 90.7 | 88.6 |
| | 10 | 98.4 | 91.0 | 89.1 |

Table 4: Ablation study on the training schedule of PolyDiffuse and RoomFormer. Results are reported in F1 scores.

| Method | Epochs | Room | Corner | Edge |
|---|---|---|---|---|
| RoomFormer | 1× | 96.2 | 88.2 | 83.9 |
| | 2× | 96.0 | 87.9 | 83.7 |
| PolyDiffuse | 1× | 98.0 | 90.6 | 88.6 |
| | 1.5× | 97.7 | 90.6 | 88.7 |
| | 2× | 97.8 | 90.7 | 88.9 |
| | 2.5× | 98.4 | 91.0 | 89.1 |

method, PolyDiffuse, outperforms the current state of the arts on floorplan [47] and HD map [23] reconstruction while enabling broader practical use cases.

**Diffusion-based generative models**: Diffusion Models (DM) or score-based generative models [28, 38, 40, 41] have made tremendous progress in the last few years and demonstrated promising performance in content generations [9, 31, 32, 33, 43] and likelihood estimations [19, 20, 28]. In this paper, we explore the potential of DM in the context of structured reconstruction and propose a Guided Set Diffusion Model (GS-DM) by extending the DDPM [14] formulation. GS-DM reconstructs complex polygonal shapes with a guided denoising (reverse) process subject to sensor data and can evaluate its reconstruction likelihood by leveraging the underlying probability flow ODE [41]. Our high-level formulation is relevant to PriorGrad [21], a diffusion model for acoustic data, but has essential differences – GS-DM learns the guidance networks to control the diffusion and direct the denoising in a per-element and stepwise manner, while PriorGrad pre-computes the mean and variance of the target Gaussian from the condition and directly shifts the diffusion/denoising trajectory.

## 7 Conclusion

This paper introduces PolyDiffuse, a system designed to reconstruct high-quality polygonal shapes from sensor data via a conditional generation procedure. At the heart of PolyDiffuse is a novel Guided Set Diffusion Model, that controls the noise injection in the diffusion process to avoid permutation ambiguity for denoising. PolyDiffuse achieves state-of-the-art performance on two challenging tasks: floorplan reconstruction from a point density image and HD map construction from onboard RGB images. To our knowledge, this paper is the first to demonstrate that Diffusion Models, generally regarded as a generation technique, are also powerful for reconstruction, potentially encouraging the community to further investigate the effectiveness of Diffusion Models in broader domains.

**Limitations**: PolyDiffuse suffers from two major limitations. First, it does not recover geometry instances (e.g., rooms for floorplan) missing in the initial reconstruction, while the likelihood-based refinement provides a possible solution by incorporating search or external inputs. Second, as shown in the HD mapping results with rough annotations, when the style of initial reconstructions (*e.g.*, fixed radius small circles) is different from the curve style of ground truth, the guidance networks could produce bad initial Gaussians and guidance, leading to inaccurate locations of the final results. Training specific guidance and denoising networks for each type of initial reconstruction is a solution but induces more computation.

**Broader impact**: The paper benefits applications in architecture, construction, autonomous driving, and potentially human-in-the-loop annotations systems to reduce human workload. However, potential malicious or unintended applications might include military scenarios, for example, reconstructing digital models of important buildings or structures for targeted military operations.

**Acknowledgements**: This research is partially supported by NSERC Discovery Grants with Accelerator Supplements and DND/NSERC Discovery Grant Supplement, NSERC Alliance Grants, and John R. Evans Leaders Fund (JELF). We thank the Digital Research Alliance of Canada and BC DRI Group for providing computational resources.

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

# Appendix:
# PolyDiffuse: Polygonal Shape Reconstruction via Guided Set Diffusion Models

This appendix is organized as follows:

- §A: The derivation of the Guided Set Diffusion Models based on the DDPM [14] formulation.
- §B: Additional implementation details, including:
  - Architectures and implementation details of the guidance networks.
  - Architectures and implementation details of the denoising networks.
  - Implementation details of the permutation loss $L_{\text{perm}}(\phi)$ for the guidance training.
  - Implementation details of the diffusion models framework.
- §C: More details and analyses on the augmented matching criterion for the mean AP metric to evaluate HD map reconstruction.
- §D: Extra experimental results including:
  - Additional ablation studies on the design choices of the denoising networks.
  - Additional qualitative results for floorplan and HD map reconstruction.

## A  Derivation of Guided Set Diffusion Models (GS-DM)

In this section, we present more details of the derivation of the Guided Set Diffusion Models (GS-DM) proposed in §3 of the main paper.

### A.1  Forward process

We derive Eq. 7, 8, and 9 in Sec. 3 of the main paper by induction. The trivial case of $t = 1$ can be easily verified. Let $\mathbf{x}_{t-1}^i = \sqrt{\bar{\alpha}_{t-1}}\mathbf{x}_0^i + \bar{\boldsymbol{\mu}}_\phi(\mathbf{x}_0, t-1, i) + \bar{\boldsymbol{\sigma}}_\phi(\mathbf{x}_0, t-1, i)\boldsymbol{\epsilon}_{t-1}^i$ for $t > 1$ and $\boldsymbol{\epsilon}_{t-1}^i \sim \mathcal{N}(\mathbf{0}, \mathbf{I})$. By definition, we have

$$
\begin{aligned}
\mathbf{x}_t^i =& \sqrt{\alpha_t}\mathbf{x}_{t-1}^i + \boldsymbol{\mu}_\phi(\mathbf{x}_0, t, i) + \sqrt{1-\alpha_t}\boldsymbol{\sigma}_\phi(\mathbf{x}_0, t, i)\boldsymbol{\epsilon}_t^i \\
=& \sqrt{\alpha_t}(\sqrt{\bar{\alpha}_{t-1}}\mathbf{x}_0^i + \bar{\boldsymbol{\mu}}_\phi(\mathbf{x}_0, t-1, i) + \bar{\boldsymbol{\sigma}}_\phi(\mathbf{x}_0, t-1, i)\boldsymbol{\epsilon}_{t-1}^i) + \\
& \boldsymbol{\mu}_\phi(\mathbf{x}_0, t, i) + \sqrt{1-\alpha_t}\boldsymbol{\sigma}_\phi(\mathbf{x}_0, t, i)\boldsymbol{\epsilon}_t^i \\
=& \sqrt{\bar{\alpha}_t}\mathbf{x}_0^i + \sqrt{\alpha_t}\bar{\boldsymbol{\mu}}_\phi(\mathbf{x}_0, t-1, i) + \boldsymbol{\mu}_\phi(\mathbf{x}_0, t, i) + \\
& \sqrt{\alpha_t}\bar{\boldsymbol{\sigma}}_\phi(\mathbf{x}_0, t-1, i)\boldsymbol{\epsilon}_t^i + \sqrt{1-\alpha_t}\boldsymbol{\sigma}_\phi(\mathbf{x}_0, t, i)\boldsymbol{\epsilon}_t^i
\end{aligned}
\tag{15}
$$

where $\boldsymbol{\epsilon}_t^i$ is a standard Gaussian variable independent of $\boldsymbol{\epsilon}_{t-1}^i$. Since $\boldsymbol{\epsilon}_{t-1}^i$ and $\boldsymbol{\epsilon}_t^i$ are independent, $\sqrt{\alpha_t}\bar{\boldsymbol{\sigma}}_\phi(\mathbf{x}_0, t-1, i)\boldsymbol{\epsilon}_{t-1}^i + \sqrt{1-\alpha_t}\boldsymbol{\sigma}_\phi(\mathbf{x}_0, t, i)\boldsymbol{\epsilon}_t^i$ is a Gaussian random variable of zero mean and a variance of $\alpha_t\bar{\boldsymbol{\sigma}}_\phi^2(\mathbf{x}_0, t-1, i) + (1-\alpha_t)\boldsymbol{\sigma}_\phi^2(\mathbf{x}_0, t, i)$. Thus we have

$$
\bar{\boldsymbol{\mu}}_\phi(\mathbf{x}_0, t, i) := \sqrt{\alpha_t}\bar{\boldsymbol{\mu}}_\phi(\mathbf{x}_0, t-1, i) + \boldsymbol{\mu}_\phi(\mathbf{x}_0, t, i) \text{ and}
\tag{16}
$$

$$
\bar{\boldsymbol{\sigma}}_\phi^2(\mathbf{x}_0, t, i) := \alpha_t\bar{\boldsymbol{\sigma}}_\phi^2(\mathbf{x}_0, t-1, i) + (1-\alpha_t)\boldsymbol{\sigma}_\phi^2(\mathbf{x}_0, t, i),
\tag{17}
$$

which correspond to Eq.8 and Eq.9 of the main paper, and we further obtain

$$
q(\mathbf{x}_t^i | \mathbf{x}_0^i) = \mathcal{N}(\mathbf{x}_t^i;\ \sqrt{\bar{\alpha}_t}\mathbf{x}_0^i + \bar{\boldsymbol{\mu}}_\phi(\mathbf{x}_0, t, i),\ \bar{\boldsymbol{\sigma}}_\phi^2(\mathbf{x}_0, t, i)\mathbf{I}),
\tag{18}
$$

which is the Eq.7 of the main paper.

### A.2  Reverse process

Assuming we are given the guidance networks $\boldsymbol{\mu}_\phi$ and $\boldsymbol{\sigma}_\phi$ and the noise scales $\boldsymbol{\sigma}_t$s of the reverse process, we follow a style similar to DDPM [14] to derive a sampling step in the reverse process. At each sampling step in the reverse process, $p_\theta(\mathbf{x}_{t-1}^i | \mathbf{x}_t^i)$, the Gaussian distribution to sample $\mathbf{x}_{t-1}^i$ from is supposed to have the same mean as $q(\mathbf{x}_{t-1}^i | \mathbf{x}_t^i, \mathbf{x}_0)$ defined by the forward process, which is also Gaussian, to minimize their KL divergence. As the joint distribution of $\mathbf{x}_{t-1}^i$ and $\mathbf{x}_t^i$ conditioned on

$\mathbf{x}_0$ is Gaussian, the mean of $q(\mathbf{x}^i_{t-1}|\mathbf{x}^i_t, \mathbf{x}_0)$ can be easily derived with the following closed form [30, Chapter 8.1.3]:

$$\sqrt{\bar{\alpha}_{t-1}}\mathbf{x}^i_0 + \bar{\boldsymbol{\mu}}_\phi(\mathbf{x}^i_0, t-1, i) + \frac{\sqrt{\alpha_t}\bar{\boldsymbol{\sigma}}^2_\phi(\mathbf{x}_0, t-1, i)}{\bar{\boldsymbol{\sigma}}^2_\phi(\mathbf{x}_0, t, i)}(\mathbf{x}^i_t - \sqrt{\bar{\alpha}_t}\mathbf{x}^i_0 - \bar{\boldsymbol{\mu}}_\phi(\mathbf{x}_0, t, i)). \quad (19)$$

Since $\mathbf{x}^i_t = \sqrt{\bar{\alpha}_t}\mathbf{x}^i_0 + \bar{\boldsymbol{\mu}}_\phi(\mathbf{x}_0, t, i) + \bar{\boldsymbol{\sigma}}_\phi(\mathbf{x}_0, t, i)\boldsymbol{\epsilon}^i$ for $\boldsymbol{\epsilon}^i \sim \mathcal{N}(\mathbf{0}, \mathbf{I})$, Equation 19 can be rewritten as

$$\sqrt{\bar{\alpha}_{t-1}}\mathbf{x}^i_0 + \bar{\boldsymbol{\mu}}_\phi(\mathbf{x}^i_0, t-1, i) + \bar{\boldsymbol{\sigma}}_\phi(\mathbf{x}_0, t, i)\frac{\sqrt{\alpha_t}\bar{\boldsymbol{\sigma}}^2_\phi(\mathbf{x}_0, t-1, i)}{\bar{\boldsymbol{\sigma}}^2_\phi(\mathbf{x}_0, t, i)}\boldsymbol{\epsilon}^i.$$

$$= \frac{1}{\sqrt{\alpha_t}}(\sqrt{\bar{\alpha}_t}\mathbf{x}^i_0 + \bar{\boldsymbol{\sigma}}_\phi(\mathbf{x}_0, t, i)\frac{\alpha_t\bar{\boldsymbol{\sigma}}^2_\phi(\mathbf{x}_0, t-1, i)}{\bar{\boldsymbol{\sigma}}^2_\phi(\mathbf{x}_0, t, i)}\boldsymbol{\epsilon}^i) + \bar{\boldsymbol{\mu}}_\phi(\mathbf{x}^i_0, t-1, i)$$

$$= \frac{1}{\sqrt{\alpha_t}}(\sqrt{\bar{\alpha}_t}\mathbf{x}^i_0 + \bar{\boldsymbol{\sigma}}_\phi(\mathbf{x}_0, t, i)\frac{\bar{\boldsymbol{\sigma}}^2_\phi(\mathbf{x}_0, t, i) - (1-\alpha_t)\boldsymbol{\sigma}^2_\phi(\mathbf{x}_0, t, i)}{\bar{\boldsymbol{\sigma}}^2_\phi(\mathbf{x}_0, t, i)}\boldsymbol{\epsilon}^i) + \bar{\boldsymbol{\mu}}_\phi(\mathbf{x}^i_0, t-1, i) \quad (20)$$

$$= \frac{1}{\sqrt{\alpha_t}}(\sqrt{\bar{\alpha}_t}\mathbf{x}^i_0 + \bar{\boldsymbol{\sigma}}_\phi(\mathbf{x}_0, t, i)\boldsymbol{\epsilon}^i + \bar{\boldsymbol{\mu}}_\phi(\mathbf{x}^i_0, t, i) - \bar{\boldsymbol{\mu}}_\phi(\mathbf{x}^i_0, t, i) - \frac{(1-\alpha_t)\boldsymbol{\sigma}^2_\phi(\mathbf{x}_0, t, i)}{\bar{\boldsymbol{\sigma}}_\phi(\mathbf{x}_0, t, i)}\boldsymbol{\epsilon}^i)$$

$$+ \bar{\boldsymbol{\mu}}_\phi(\mathbf{x}^i_0, t-1, i)$$

$$= \frac{1}{\sqrt{\alpha_t}}(\mathbf{x}^i_t - \bar{\boldsymbol{\mu}}_\phi(\mathbf{x}^i_0, t, i) - \frac{(1-\alpha_t)\boldsymbol{\sigma}^2_\phi(\mathbf{x}_0, t, i)}{\bar{\boldsymbol{\sigma}}_\phi(\mathbf{x}_0, t, i)}\boldsymbol{\epsilon}^i) + \bar{\boldsymbol{\mu}}_\phi(\mathbf{x}^i_0, t-1, i).$$

As our formulation directly parameterizes $\bar{\sigma}_\phi$, we further simplify Eq. 20 by defining $\bar{\sigma}_\phi(\mathbf{x}_0, t, i) := \sqrt{1-\bar{\alpha}_t}C(\mathbf{x}_0, i)$ where $C(\mathbf{x}_0, i)$ is independent of the timestep $t$, thus implicitly defining $\boldsymbol{\sigma}_\phi(\mathbf{x}_0, t, i) = \bar{\boldsymbol{\sigma}}_\phi(\mathbf{x}_0, T, i) = C(\mathbf{x}_0, i)$. Such a parameterization makes $\{\bar{\boldsymbol{\sigma}}_\phi(\mathbf{x}_0, t, i)\}_t$ simply an interpolation between 0 and $\bar{\boldsymbol{\sigma}}_\phi(\mathbf{x}_0, T, i)$ and further simplifies Eq. 20 as

$$\frac{1}{\sqrt{\alpha_t}}(\mathbf{x}^i_t - \bar{\boldsymbol{\mu}}_\phi(\mathbf{x}^i_0, t, i) - \bar{\boldsymbol{\sigma}}_\phi(\mathbf{x}_0, t, i)\frac{1-\alpha_t}{1-\bar{\alpha}_t}\boldsymbol{\epsilon}^i) + \bar{\boldsymbol{\mu}}_\phi(\mathbf{x}^i_0, t-1, i). \quad (21)$$

During inference, we replace $\bar{\boldsymbol{\mu}}_\phi(\mathbf{x}_0, t, i)$ and $\bar{\boldsymbol{\sigma}}_\phi(\mathbf{x}_0, t, i)$ with $\bar{\boldsymbol{\mu}}_\phi(\hat{\mathbf{x}}_0, t, i)$ and $\bar{\boldsymbol{\sigma}}_\phi(\hat{\mathbf{x}}_0, t, i)$ respectively and replace $\boldsymbol{\epsilon}^i$ with the denoising network $\boldsymbol{\epsilon}^i_\theta$ to define the mean of sampling distribution $p_\theta(\mathbf{x}^i_{t-1}|\mathbf{x}^i_t)$ and derive a sampling step in the reverse process as

$$\mathbf{x}^i_{t-1} = \frac{1}{\sqrt{\alpha_t}}\left[\mathbf{x}^i_t - \bar{\boldsymbol{\mu}}_\phi(\hat{\mathbf{x}}_0, t, i) - \bar{\boldsymbol{\sigma}}_\phi(\hat{\mathbf{x}}_0, t, i)\frac{1-\alpha_t}{1-\bar{\alpha}_t}\boldsymbol{\epsilon}^i_\theta(\mathbf{x}_t, t, \mathbf{y})\right] + \bar{\boldsymbol{\mu}}_\phi(\hat{\mathbf{x}}_0, t-1, i) + \boldsymbol{\sigma}_t\mathbf{z}^i, \quad (22)$$

which is the Eq.10 of the main paper.

# B  Additional implementation details

In this section, we complement §4 of the main paper by providing the complete implementation details of PolyDiffuse for the two different tasks.

## B.1  Guidance networks

The implementation details of the guidance networks are shared across the two tasks.

**Model architectures**: In §4 of the main paper, we parameterize $\bar{\boldsymbol{\mu}}_\phi(\mathbf{x}_0, T, i)$ and $\bar{\boldsymbol{\sigma}}_\phi(\mathbf{x}_0, T, i)$ with two Transformers $\text{Trans}_{\bar{\boldsymbol{\mu}}_\phi}(\mathbf{x}_0, i)$ and $\text{Trans}_{\bar{\boldsymbol{\sigma}}_\phi}(\mathbf{x}_0, i)$, and define $\bar{\boldsymbol{\mu}}_\phi(\mathbf{x}_0, t, i)$ and $\bar{\boldsymbol{\sigma}}_\phi(\mathbf{x}_0, t, i)$ with Eq.14. The two Transformers are implemented with a DETR-style [3] Transformer decoder[2]: The Transformer decoder contains two shared attention-based decoder layers and two separate linear projection heads for $\bar{\boldsymbol{\mu}}_\phi$ and $\bar{\boldsymbol{\sigma}}_\phi$, respectively. Each decoder layer consists of an intra-element self-attention layer and a global self-attention layer, whose outputs are fused by addition. Each vertex in the input $\mathbf{x}_0$ becomes an input node of the Transformer decoder, and the node feature consists of three positional encodings [44]: 1) positional encoding of the X-axis coordinate, 2) positional

---

[2] https://github.com/facebookresearch/detr/blob/main/models/transformer.py

encoding of the Y-axis coordinate, 3) positional encoding of the vertex index inside the element. The sequence of each element starts with a special dummy node (similar to the SOS token in language modeling), whose final encodings are used to compute the $\bar{\mu}_\phi$ and $\bar{\sigma}_\phi$ of the element. The dimension of the positional encoding is 128, and the hidden dimension of the Transformer decoder is 256.

**Training details**: The guidance training is summarized by Algorithm 1 of the main paper. We train the Transformer decoder for 3125 iterations with batch size 32. Adam optimizer is employed with a learning rate of 2e-4 and a weight decay rate of 1e-4.

## B.2 Denoising networks

Since the implementations of the denoising networks are based on the corresponding state-of-the-art task-specific models, we describe them separately.

**Floorplan reconstruction**: The denoising network for the floorplan reconstruction task refers to RoomFormer [47] and its official implementation[3]. RoomFormer is a DETR-based [3] model consisting of a ResNet50 [12] image backbone, a Transformer encoder to process and aggregate image information, and a Transformer decoder with two-level learnable embeddings/cooridnates to predict a set of polygon vertices from the image information. The Transformer has 6 encoder layers and 6 decoder layers with an embedding dimension of 256, and deformable-attention [50] is employed for all cross-attention layers and the self-attention layers in the Transformer encoder.

We follow the same architecture as RoomFormer, and have discussed the key modifications to turn the model into a denoising function (§4 of the main paper) and our improvements to lift up its overall performance (§5 of the main paper). A minor modification omitted in the main paper is that we add an intra-element attention layer to each Transformer decoder layer to increase the model capacity, as we found the model converges obviously slower under the denoising formulation than the regression/detection formulation of RoomFormer. In each decoder layer, the outputs of the intra-element attention are added to the outputs of the original global attention. The intra-element attention increases the number of overall parameters by ~5%, and the corresponding ablation study is in §D.1 of this appendix, showing minor but recognizable improvements.

For training the denoising network, we keep the same setups as RoomFormer except that we increase the number of training iterations (ablation study is in §5 of the main paper). Adam optimizer is employed with a base learning rate of 2e-4 for all parameters, and the learning rate decays by a factor of 0.1 for the last 20% iterations.

**HD map reconstruction**: The denoising network for the HD map reconstruction task refers to MapTR [23] and its official implementation[4]. The overall design of MapTR is very similar to RoomFormer, where the keys are the "hierarchical" or "two-level" query embeddings for DETR-style Transformer and a loss based on "hierarchical bipartite matching". We use the same model config file provided in MapTR's official codebase and convert the model into a denoising function as described in §4 of the main paper. Similar to what we did to RoomFormer above, we add an intra-element attention layer to each Transformer decoder layer to facilitate convergence.

We employ an Adam optimizer with a base learning rate of 6e-4 and a weight decay factor of 1e-4. A cosine learning rate scheduler is used. To further facilitate convergence under the denoising formulation, we load the ResNet image backbone and Transformer encoder from the pre-trained MapTR, and set the initial learning rate of the image backbone to be 0.1 of the base learning rate.

When using MapTR as the proposal generator to produce the initial reconstruction for PolyDiffuse, we only consider predicted instances with a confidence score higher than $0.5$ as MapTR's positive predictions. PolyDiffuse takes and updates the positive predictions of MapTR and keeps the remaining low-confidence predictions unchanged. Only the positive predictions are visualized for the qualitative results of MapTR and PolyDiffuse.

---

[3]https://github.com/ywyue/RoomFormer
[4]https://github.com/hustvl/MapTR

## B.3 Permutation loss

Directly computing the $L_{\text{perm}}(\phi)$ as defined in Eq.14 of the main paper requires finding $\mathbf{x}_0^* = \arg\max_{\mathbf{x}_0' \in \mathcal{P}^!(\mathbf{x}_0)\setminus\{\mathbf{x}_0\}} L_{\text{Triplet}}(\mathbf{x}_t, \mathbf{x}_0, \mathbf{x}_0')$ and $\mathbf{x}_t^* = \arg\max_{\mathbf{x}_t' \in \mathcal{P}^!(\mathbf{x}_t)\setminus\{\mathbf{x}_t\}} L_{\text{Triplet}}(\mathbf{x}_0, \mathbf{x}_t, \mathbf{x}_t')$, where the size of $\mathcal{P}^!(\mathbf{x}_0)$ and $\mathcal{P}^!(\mathbf{x}_t)$ is $N!$. This enumeration over all $N!$ permutations of $\mathbf{x}_0$ and $\mathbf{x}_t$ immediately becomes computationally prohibitive when $N$ gets large.

To reduce the computational cost and make Eq.14 practically feasible, we propose to approximate $L_{\text{perm}}(\phi)$ with element-level triplet losses. Concretely, we first enumerate all pairs of elements $\mathbf{x}_0^i$ and $\mathbf{x}_t^j$ for $i, j = 1, \ldots, N$ to compute a $N \times N$ distance matrix $D$. The entry $D(i, j)$ is the minimum distance between two elements $\mathbf{x}_0^i$ and $\mathbf{x}_t^j$, considering all possible vertex-level permutations (*i.e.*, two variants for a polyline, and $2(N_i - 1)$ variants for a polygon with $N_i$ vertices, similar to the "point-level matching" in MapTR [23]). We then replace the sample-level triplet loss in $L_{\text{perm}}(\phi)$ with $N$ element-level triplet losses, and define the computationally feasible proxy loss $\hat{L}_{\text{perm}}(\phi)$ as:

$$\hat{L}_{\text{perm}}(\phi) = \sum_{i=1,\ldots,N} \max_{1 \leq j \leq N, j \neq i} L_{\text{Triplet}}(\mathbf{x}_t^i, \mathbf{x}_0^i, \mathbf{x}_0^j) + \max_{1 \leq j \leq N, j \neq i} L_{\text{Triplet}}(\mathbf{x}_0^i, \mathbf{x}_t^i, \mathbf{x}_t^j), \quad (23)$$

$$L_{\text{Triplet}}(\mathbf{x}_t^i, \mathbf{x}_0^i, \mathbf{x}_0^j) = \max\left(0, \alpha + D(i, i) - D(j, i)\right), \quad (24)$$

$$L_{\text{Triplet}}(\mathbf{x}_0^i, \mathbf{x}_t^i, \mathbf{x}_t^j) = \max\left(0, \alpha + D(i, i) - D(i, j)\right). \quad (25)$$

$\alpha$ is the soft margin hyperparameter of the hinge-style triplet loss [15, 34] and we set $\alpha = 0.1$. All coordinate values are re-scaled into $[-1, 1]$. In this way, we reduce the computational cost from $O(N!)$ to $O(N^2 M)$, where $M = \max_{i=1}^{N} N_i$ is the maximum number of vertices of an element of $\mathbf{x}_0$. In practical implementation, the guidance training (Algorithm 1 of the main paper) uses $\hat{L}_{\text{perm}}(\phi)$ rather than $L_{\text{perm}}(\phi)$.

## B.4 Diffusion models framework

Karras et al.[19] (EDM) presents a general diffusion model framework, where DDPM [14] and SDE-based DM formulations from Song et al.[41] can all be viewed as specializations of the proposed framework. We borrow its official codebase[5] to implement our GS-DM as it provides a general and clean base implementation suitable for all DM-based formulations. We then describe how we adapt the GS-DM into the EDM-based framework and list the relevant hyperparameter settings. This subsection follows the notations in Karras et al., where $\boldsymbol{y}$ is the data sample, $\boldsymbol{n}$ is the sampled Gaussian noise, $\boldsymbol{x}$ is the noisy sample, $\sigma$ is the noise level (equivalent to the timestep), while $c_{\text{skip}}(\sigma)$, $c_{\text{in}}(\sigma)$, $c_{\text{out}}(\sigma)$, and $c_{\text{noise}}(\sigma)$ are the preconditioning factors [19, Section 5]. Similar to the notations of our main paper, we let $\boldsymbol{y}^i$ and $\boldsymbol{x}^i$ denote the $i^{\text{th}}$ element of $\boldsymbol{y}$ and $\boldsymbol{x}$, respectively. The sensor condition is omitted for notation simplicity.

To adapt our GS-DM into the EDM framework, we first set $\sigma_{\text{data}} = 1.0$ for EDM [19, Table 1], and then adapt the preconditioning equation [19, Section 5, Eq.7] based on §3 of our main paper:

$$D_\theta(\boldsymbol{x}^i; \sigma, \boldsymbol{y}) = c_{\text{skip}}(\sigma)\, \boldsymbol{x}^i + c_{\text{out}}(\sigma)\, F_\theta^i\left(\left\{c_{\text{in}}(\sigma)\, \boldsymbol{x}^i + (1 - c_{\text{in}}(\sigma))\, \bar{\boldsymbol{\mu}}_\phi(\boldsymbol{y}, \sigma, i)\right\}; c_{\text{noise}}(\sigma)\right), \quad (26)$$

where the per-element noise injection is defined as $\boldsymbol{x}^i = \boldsymbol{y}^i + \boldsymbol{n}\bar{\boldsymbol{\sigma}}_\phi(\boldsymbol{y}, \sigma, i)$, and noise $\boldsymbol{n} \sim \mathcal{N}(\boldsymbol{0}, \sigma^2 \mathbf{I})$. $D_\theta(\boldsymbol{x}^i; \sigma, \boldsymbol{y})$ is the reconstructed $\boldsymbol{y}^i$. $F_\theta$ is the denoising network taking all elements of a noisy sample $\boldsymbol{x}$, and $F_\theta^i$ is the output of the $i^{\text{th}}$ element. With the above modifications, we implement our GS-DM with the general EDM framework. We then describe the concrete hyperparameter settings [19, Table 1] for our two tasks. Please refer to Karras et al. for detailed explanations of each hyperparameter.

**Floorplan reconstruction**: For the guidance training, we set the $P_{\text{mean}} = 1.0$ and $P_{\text{std}} = 4.0$ to ensure sufficient coverage of the forward process. For the denoising training, we set the $P_{\text{mean}} = -0.5$ and $P_{\text{std}} = 1.5$. For inference (sampling), we set $\sigma_{\text{max}} = 5.0$ and $\sigma_{\text{min}} = 0.01$. Instead of the $2^{\text{nd}}$ order Heun ODE solver adopted by the original EDM, we simply employ the $1^{\text{st}}$ order Euler solver for sampling, which is equivalent to DDIM [39]. All other hyperparameters are kept unchanged.

**HD map reconstruction**: The settings of $P_{\text{mean}}$ and $P_{\text{std}}$ for guidance and denoising training are the same as the floorplan reconstruction task. We set $c_{\text{skip}}(\sigma) = 0$ and $c_{\text{out}}(\sigma) = 1$ so that the denoising

---

[5] https://github.com/NVlabs/edm

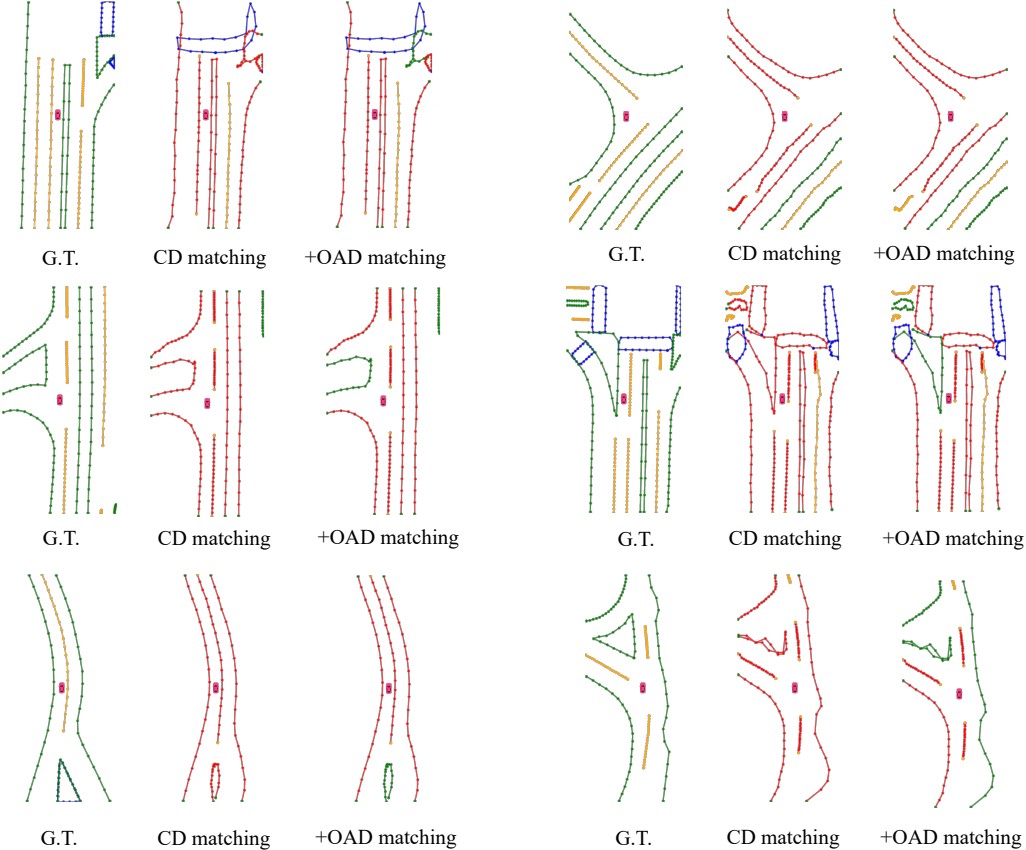

Figure 6: Illustration of the matching results with MapTR [23] predictions. **CD matching**: the original Chamfer distance (CD) matching criterion with a threshold of $1.0m$. **+OAD matching**: the original CD criterion with a threshold of $1.0m$, augmented with the order-aware angle distance (OAD) criterion with a threshold of $10°$. The matched instances (*i.e.*, true positives) are marked in red except for the two endpoints.

network directly estimates the sample $y$, as we found this choice stabilizes the denoising training on the HD map construction task. For inference (sampling), we set $\sigma_{max} = 5.0$ and $\sigma_{min} = 0.1$, and use the Euler solver. All other hyperparameters are kept unchanged.

## C   The augmented matching criterion for HD map reconstruction

In §5.2 of the main paper, we augment the Chamfer-distance (CD) matching criterion for the mean AP (mAP) metric used in previous works [23, 25] with an order-aware angle distance (OAD) to better evaluate the structural regularity and directional correctness of the results. This section first discusses the limitations of the original Chamfer-distance matching criterion and then provides complete implementation details of the order-aware angle distance.

### C.1   The limitations of the Chamfer-distance matching criterion

Figure 6 shows examples of the Chamfer-distance (CD) matching (threshold is $1.0m$). Since CD considers each vertex separately, it mostly evaluates the global location of the instance and ignores the structural/directional information of the prediction. In the figure, some predictions with obviously wrong shapes are considered true positives under the CD matching criterion, while the augmented OAD matching criterion can effectively reject these bad shapes and align better with human judgment.

## C.2 Implementation details

The computation of the order-aware angle distance is based on an optimal vertex-level matching between a predicted element (instance) and a ground truth (G.T.) element, similar to the "point-level matching" in the loss computation of MapTR [23]. Concretely, we enumerate through all equivalent representations of a G.T. map element and compute the average vertex-level $L1$ distance between the G.T. vertices and the predicted vertices. The variant with the smallest average vertex-level $L1$ distance forms the optimal matching with the predicted element. There are two equivalent variants for a polyline and $2(N_i - 1)$ equivalent variants for a polygon with $N_i$ vertices. After obtaining the optimal one-to-one vertex-level matching, we trace along the vertices of the G.T. and predicted elements to compute their average angle distance.

With the augmented OAD matching criterion, we change the three-level thresholds of the original CD-based AP from $\{0.5m, 1.0m, 1.5m\}$ into $\{(0.5m, 5°), (1.0m, 10°), (1.5m, 15°)\}$. However, compared to polylines (*i.e.*, road dividers and boundaries), we noticed that the angle direction of polygons (pedestrian crossings) is much more challenging to recover. With an OAD threshold of $5°$, MapTR's average precision for the pedestrian crossing class ($AP_p$) is zero. Therefore, we loosen the OAD thresholds for the pedestrian crossing class by a factor of 2 (*i.e.*, $\{(0.5m, 10°), (1.0m, 20°), (1.5m, 30°)\}$) while not changing the thresholds for the other two classes.

## D  Additional experimental results

This section provides additional experimental results, including extra ablation studies and qualitative comparisons.

Table 5: Ablation study for the intra-element attention layer on the floorplan reconstruction task. Note that PolyDiffuse uses the RoomFormer from the first row to produce the initial reconstruction.

| Evaluation Level → | | Room | | | Corner | | | Angle | | |
|---|---|---|---|---|---|---|---|---|---|---|
| Method | Intra-element-attn | Prec. | Rec. | F1 | Prec. | Rec. | F1 | Prec. | Rec. | F1 |
| RoomFormer | ✗ | 96.3 | 96.2 | 96.2 | 89.7 | 86.7 | 88.2 | 85.4 | 82.5 | 83.9 |
| RoomFormer | ✓ | 96.9 | 96.4 | 96.6 | 90.3 | 87.0 | 88.6 | 84.9 | 81.9 | 83.4 |
| PolyDiffuse(Ours) | ✗ | 98.4 | 97.8 | 98.1 | 92.4 | 89.0 | 90.7 | 90.2 | 87.0 | 88.6 |
| PolyDiffuse(Ours) | ✓ | 98.7 | 98.1 | 98.4 | 92.8 | 89.3 | 91.0 | 90.8 | 87.4 | 89.1 |

Table 6: Ablation study for the intra-element attention layer on the HD map reconstruction task. Note that PolyDiffuse uses the MapTR from the first row to produce the initial reconstruction.

| Matching Criterion → | | Chamfer distance | | | | + Ordered angle distance | | | |
|---|---|---|---|---|---|---|---|---|---|
| Method | Intra-element-attn | $AP_p$ | $AP_d$ | $AP_b$ | mAP | $AP_p$ | $AP_d$ | $AP_b$ | mAP |
| MapTR | ✗ | 55.8 | 60.9 | 61.1 | 59.3 | 46.1 | 43.4 | 41.9 | 43.8 |
| MapTR | ✓ | 56.5 | 59.9 | 60.2 | 58.8 | 48.6 | 44.8 | 41.6 | 45.0 |
| PolyDiffuse(Ours) | ✗ | 56.9 | 59.2 | 60.6 | 58.9 | 50.8 | 48.3 | 44.9 | 48.0 |
| PolyDiffuse(Ours) | ✓ | 58.2 | 59.7 | 61.3 | 59.7 | 52.0 | 49.5 | 45.4 | 49.0 |

### D.1  Additional ablation studies

Table 5 presents the ablation study of the intra-element attention with the floorplan reconstruction task. Comparing the first and second rows, adding intra-element attention to RoomFormer slightly improves the room and corner results but deteriorates the angle-level performance. Table 6 provides a similar comparison for the HD map reconstruction task. Augmenting MapTR with intra-element attention worsens the mAP with the CD matching criterion while slightly improving the mAP with the OAD-augmented matching criterion. These comparisons provide a potential empirical explanation for the architecture choice of RoomFormer and MapTR – they only employ global attention layers without extra attention layers at the element or instance level.

On the contrary, as indicated by the third and fourth rows of both Table 5 and Table 6, applying intra-element attention consistently boosts PolyDiffuse across all metrics, although the improvements are not huge. A potential explanation here is that the denoising task of PolyDiffuse is more challenging than the regression/detection task of RoomFormer and MapTR, thus benefiting from extra modeling capacities.

## D.2   A toy experiment with standard DM

In Figure 7, we provide a toy experiment to support what we motivated in §1 of the main paper, which demonstrates how standard DM easily fails even with a single data sample. Note that we have clarified the definition of standard DM in §5.3 of the main paper. In this experiment, the data contains a single toy sample with 6 rectangular shapes, so there are permutation-equivalent representations. After sufficient training, we draw four samples using the image-conditioned denoising process. The DDIM sampler is used with 10 sampling steps, so the randomness only comes from the initial noise. As the figure shows, only the third sample gets the correct reconstruction result. With the challenges of set ambiguity, a standard conditional DM has trouble overfitting a single data sample and easily gets wrong outputs when the initial noise is inappropriate.

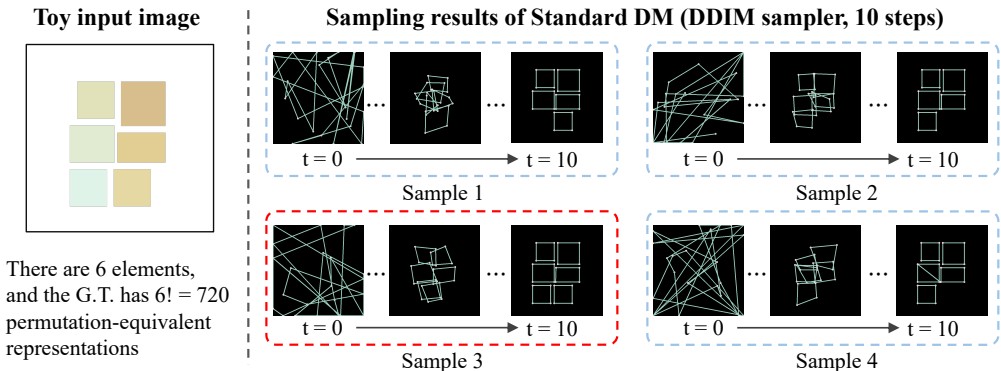

Figure 7: A simple toy experiment of using a standard DM to fit a single data sample with 6 elements. Four sampling results with *different initial noises* are shown. The DDIM sampler is used with 10 denoising steps. Only one of the four samples (*i.e.*, Sample 3) gets the correct final result due to the challenges induced by the set ambiguity, as explained in the main paper.

## D.3   More qualitative examples

We provide additional qualitative results to compare PolyDiffuse against the state-of-the-art floorplan and HD map reconstruction methods, respectively. Note that the state-of-the-art method (*i.e.*, Room-Former or MapTR) produces the initial reconstruction for PolyDiffuse. Figure 8 to Figure 10 are for the floorplan reconstruction task, while Figure 11 to Figure 13 are for the HD map reconstruction task.

Since the likelihood-based refinement is not employed in the qualitative examples of this subsection, PolyDiffuse cannot discover missing instances that are not covered by the proposal generator (*i.e.*, RoomFormer and MapTR in the qualitative results). However, the visual comparisons clearly demonstrate that PolyDiffuse significantly improves the structural regularity of the reconstructed polygonal shapes.

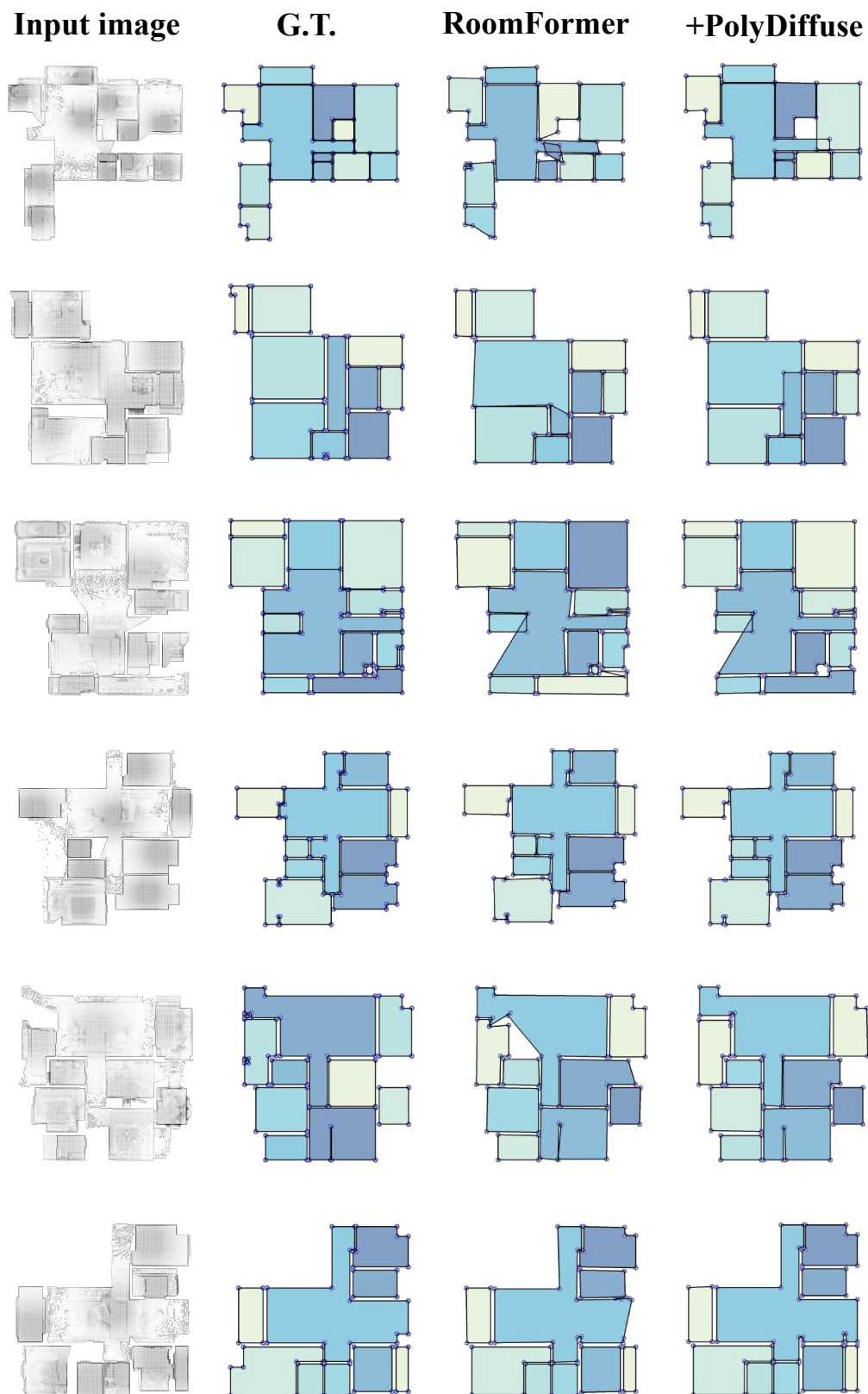

Figure 8: Additional qualitative results for the floorplan reconstruction task.

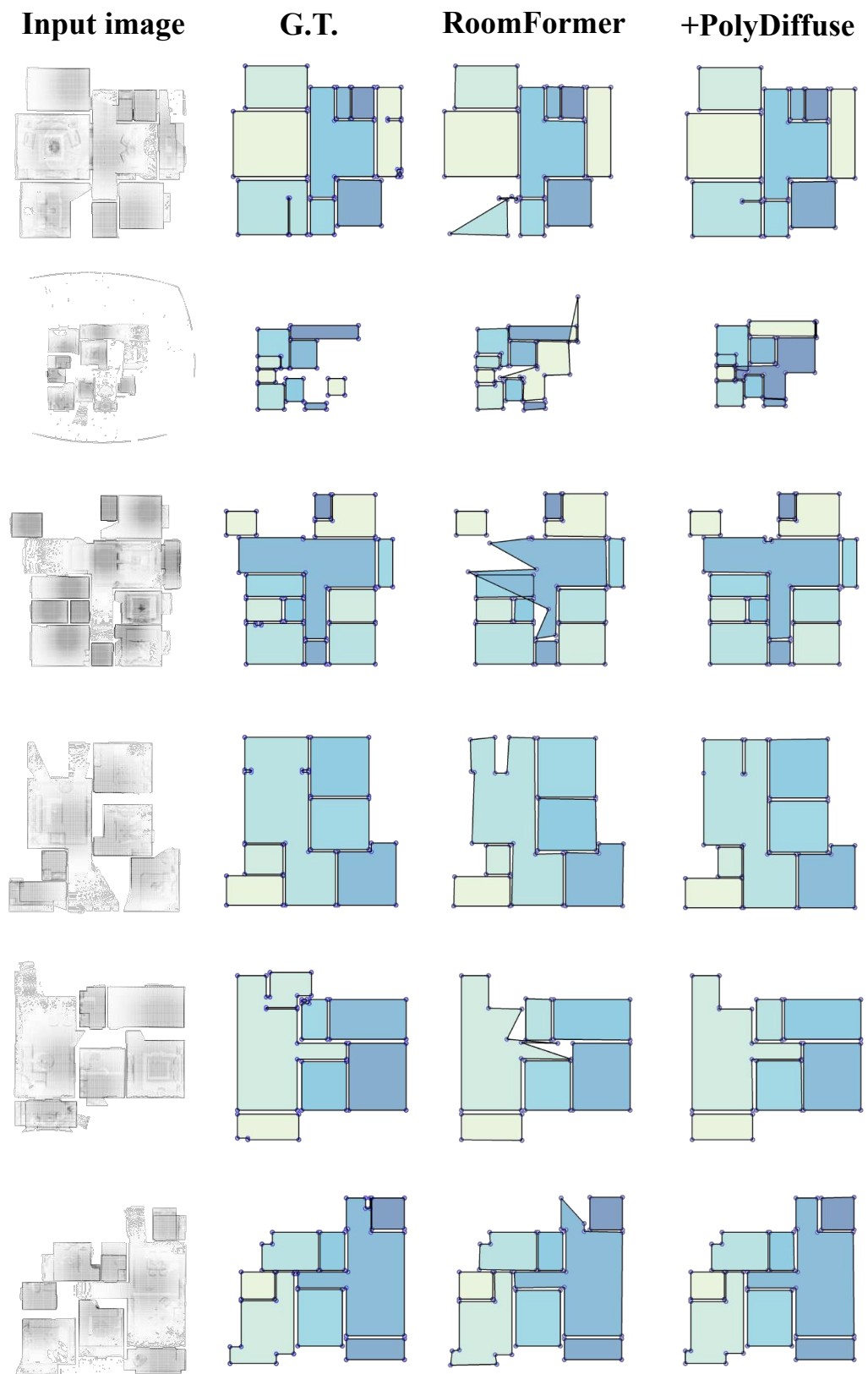

Figure 9: Additional qualitative results for the floorplan reconstruction task.

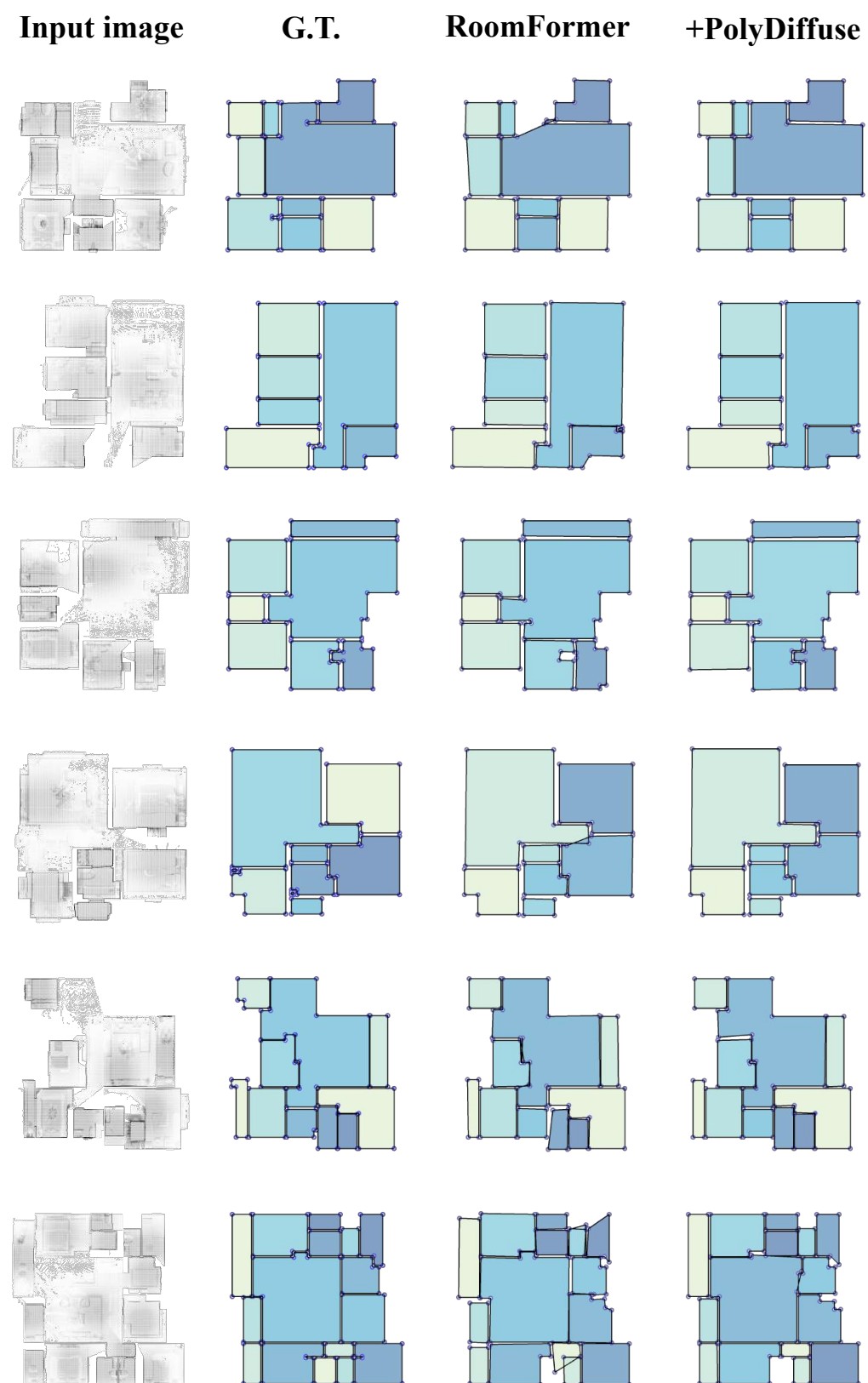

Figure 10: Additional qualitative results for the floorplan reconstruction task.

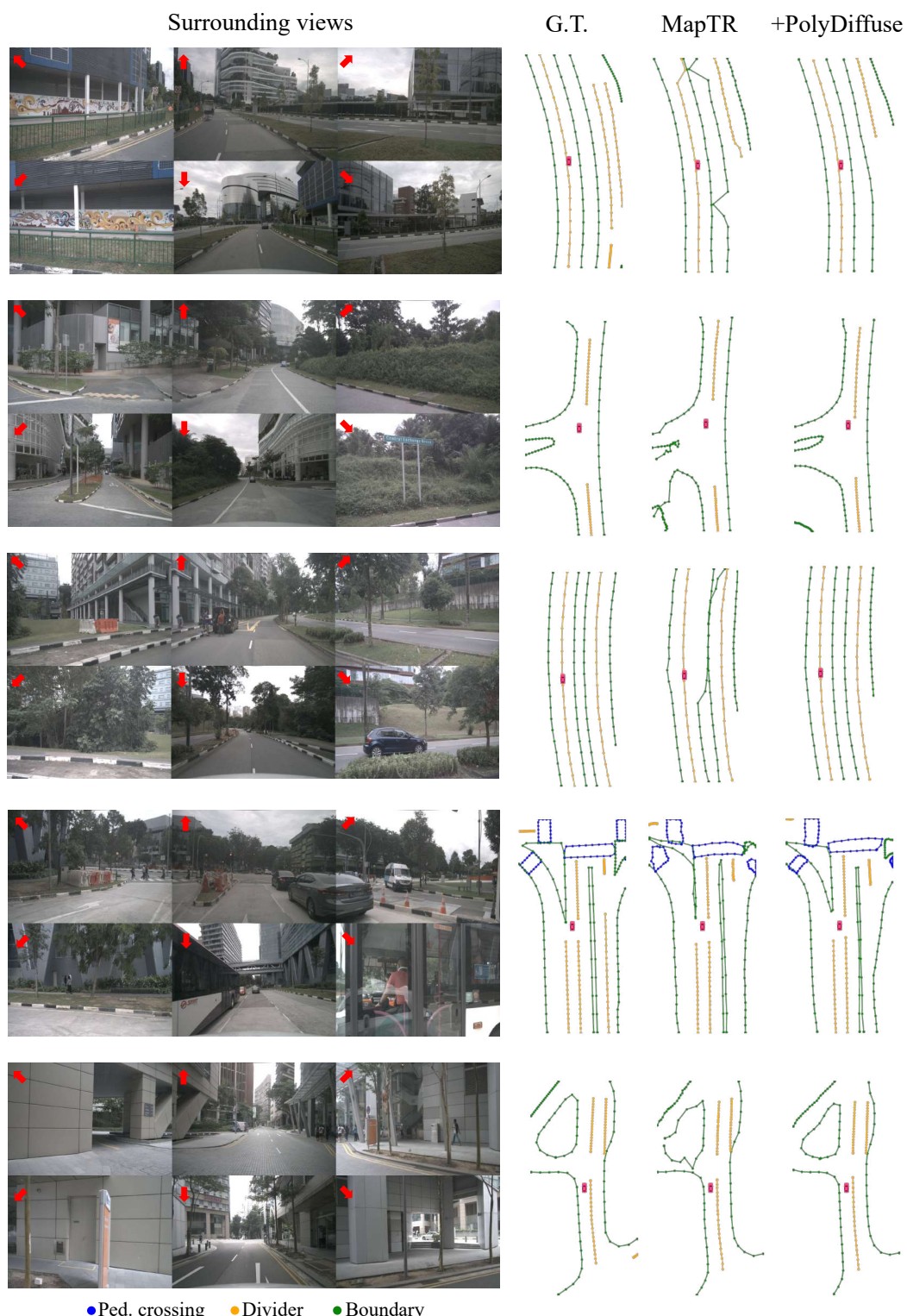

Figure 11: Additional qualitative results of the HD map reconstruction task.

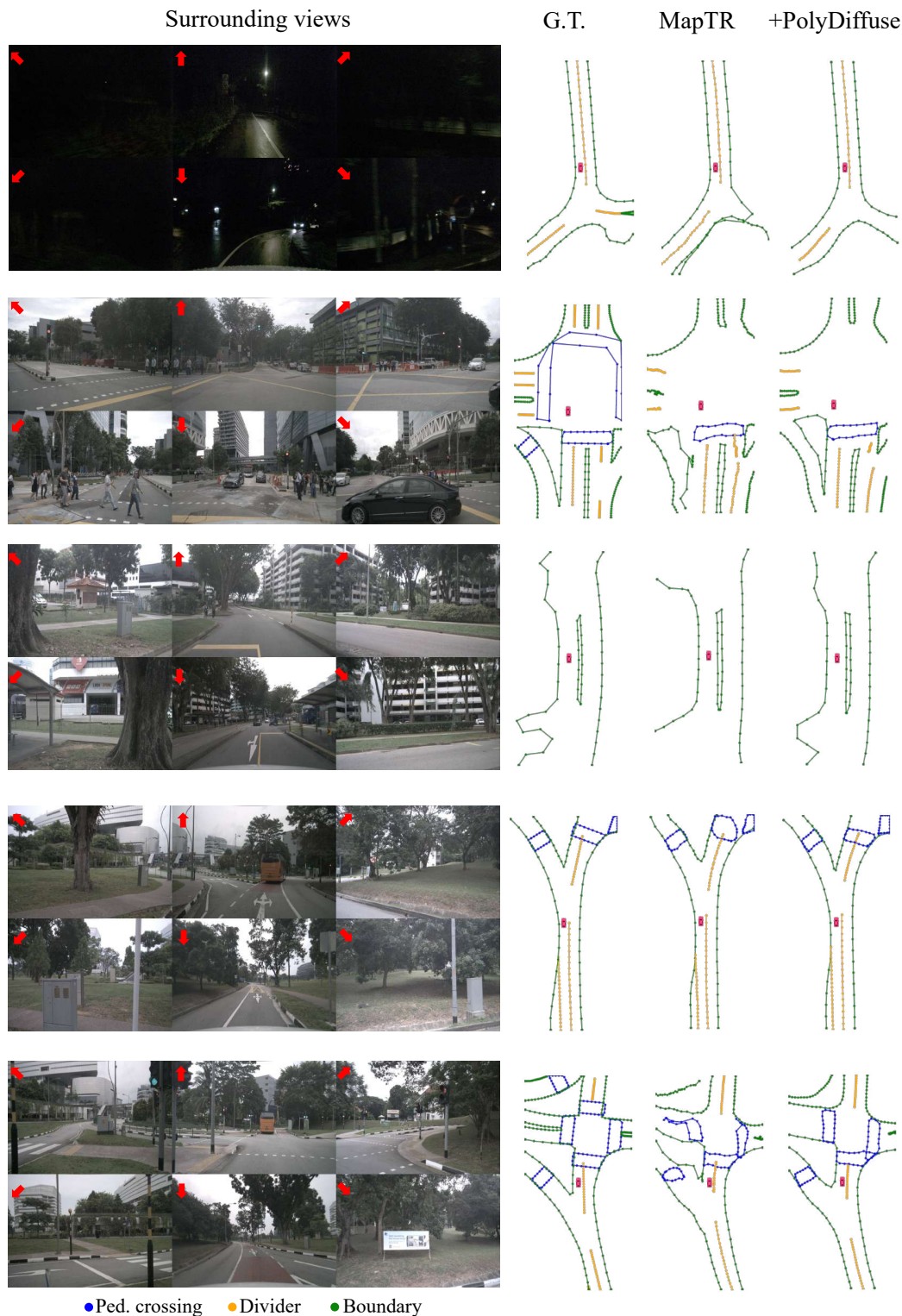

Figure 12: Additional qualitative results of the HD map reconstruction task.

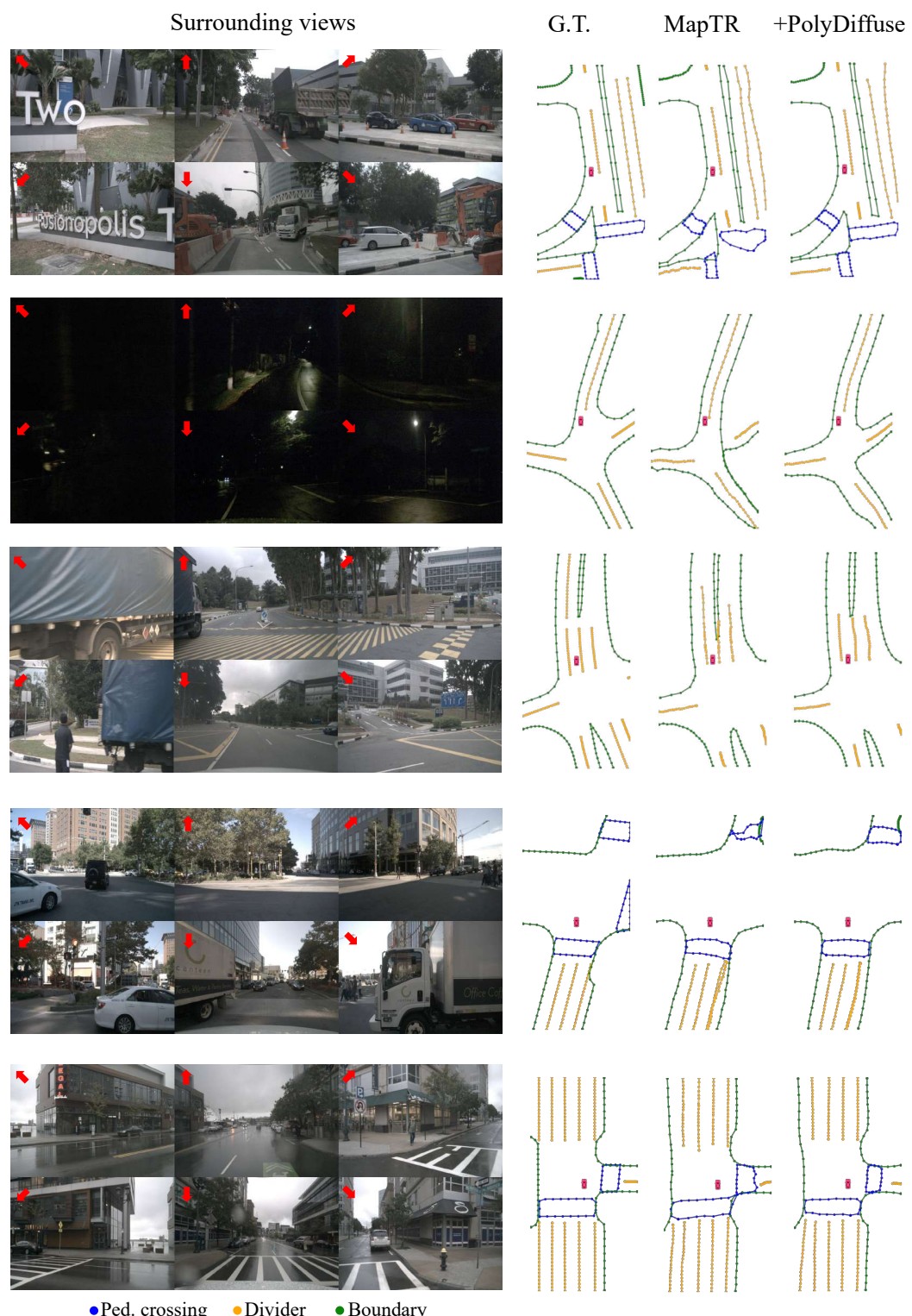

Surrounding views      G.T.     MapTR    +PolyDiffuse

● Ped. crossing    ● Divider    ● Boundary

Figure 13: Additional qualitative results of the HD map reconstruction task.

