# Supplementary Document:

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

$$\hat{L}_{\mathrm{perm}}(\phi) = \sum_{i=1,\ldots,N} \max_{1 \le j \le N, j \neq i} L_{\mathrm{Triplet}}(\mathbf{x}_t^i, \mathbf{x}_0^i, \mathbf{x}_0^j) + \max_{1 \le j \le N, j \neq i} L_{\mathrm{Triplet}}(\mathbf{x}_0^i, \mathbf{x}_t^i, \mathbf{x}_t^j), \quad (9)$$

$$L_{\mathrm{Triplet}}(\mathbf{x}_t^i, \mathbf{x}_0^i, \mathbf{x}_0^j) = \max\left(0, \alpha + D(i, i) - D(j, i)\right), \quad (10)$$

$$L_{\mathrm{Triplet}}(\mathbf{x}_0^i, \mathbf{x}_t^i, \mathbf{x}_t^j) = \max\left(0, \alpha + D(i, i) - D(i, j)\right). \quad (11)$$

$\alpha$ is the soft margin hyperparameter of the hinge-style triplet loss [5, 10] and we set $\alpha = 0.1$. All coordinate values are re-scaled into $[-1, 1]$. In this way, we reduce the computational cost from $O(N!)$ to $O(N^2 M)$, where $M = \max_{i=1}^N N_i$ is the maximum number of vertices of an element of $\mathbf{x}_0$. In practical implementation, the guidance training (Algorithm 1 of the main paper) uses $\hat{L}_{\mathrm{perm}}(\phi)$ rather than $L_{\mathrm{perm}}(\phi)$.

## 2.4 Diffusion models framework

Karras et al.[6] (EDM) presents a general diffusion model framework, where DDPM [4] and SDE-based DM formulations from Song et al.[12] can all be viewed as specializations of the proposed framework. We borrow its official codebase[4] to implement our GS-DM as it provides a general and clean base implementation suitable for all DM-based formulations. We then describe how we adapt the GS-DM into the EDM-based framework and list the relevant hyperparameter settings. This subsection follows the notations in Karras et al., where $\boldsymbol{y}$ is the data sample, $\boldsymbol{n}$ is the sampled Gaussian noise, $\boldsymbol{x}$ is the noisy sample, $\sigma$ is the noise level (equivalent to the timestep), while $c_{\mathrm{skip}}(\sigma)$, $c_{\mathrm{in}}(\sigma)$, $c_{\mathrm{out}}(\sigma)$, and $c_{\mathrm{noise}}(\sigma)$ are the preconditioning factors [6, Section 5]. Similar to the notations of our main paper, we let $\boldsymbol{y}^i$ and $\boldsymbol{x}^i$ denote the $i^{\mathrm{th}}$ element of $\boldsymbol{y}$ and $\boldsymbol{x}$, respectively. The sensor condition is omitted for notation simplicity.

To adapt our GS-DM into the EDM framework, we first set $\sigma_{\mathrm{data}} = 1.0$ for EDM [6, Table 1], and then adapt the preconditioning equation [6, Section 5, Eq.7] based on §3 of our main paper:

$$D_\theta(\boldsymbol{x}^i; \sigma, \boldsymbol{y}) = c_{\mathrm{skip}}(\sigma)\, \boldsymbol{x}^i + c_{\mathrm{out}}(\sigma)\, F_\theta^i\left(\left\{c_{\mathrm{in}}(\sigma)\, \boldsymbol{x}^i + (1 - c_{\mathrm{in}}(\sigma))\, \bar{\boldsymbol{\mu}}_\phi(\boldsymbol{y}, \sigma, i)\right\};\, c_{\mathrm{noise}}(\sigma)\right), \quad (12)$$

where the per-element noise injection is defined as $\boldsymbol{x}^i = \boldsymbol{y}^i + \boldsymbol{n}\bar{\boldsymbol{\sigma}}_\phi(\boldsymbol{y}, \sigma, i)$, and noise $\boldsymbol{n} \sim \mathcal{N}(\boldsymbol{0}, \sigma^2 \mathbf{I})$. $D_\theta(\boldsymbol{x}^i; \sigma, \boldsymbol{y})$ is the reconstructed $\boldsymbol{y}^i$. $F_\theta$ is the denoising network taking all elements of a noisy sample $\boldsymbol{x}$, and $F_\theta^i$ is the output of the $i^{\mathrm{th}}$ element. With the above modifications, we implement our GS-DM with the general EDM framework. We then describe the concrete hyperparameter settings [6, Table 1] for our two tasks. Please refer to Karras et al. for detailed explanations of each hyperparameter.

**Floorplan reconstruction**: For the guidance training, we set the $P_{\mathrm{mean}} = 1.0$ and $P_{\mathrm{std}} = 4.0$ to ensure sufficient coverage of the forward process. For the denoising training, we set the $P_{\mathrm{mean}} = -0.5$ and $P_{\mathrm{std}} = 1.5$. For inference (sampling), we set $\sigma_{\mathrm{max}} = 5.0$ and $\sigma_{\mathrm{min}} = 0.01$. Instead of the $2^{\mathrm{nd}}$

---

[4] https://github.com/NVlabs/edm

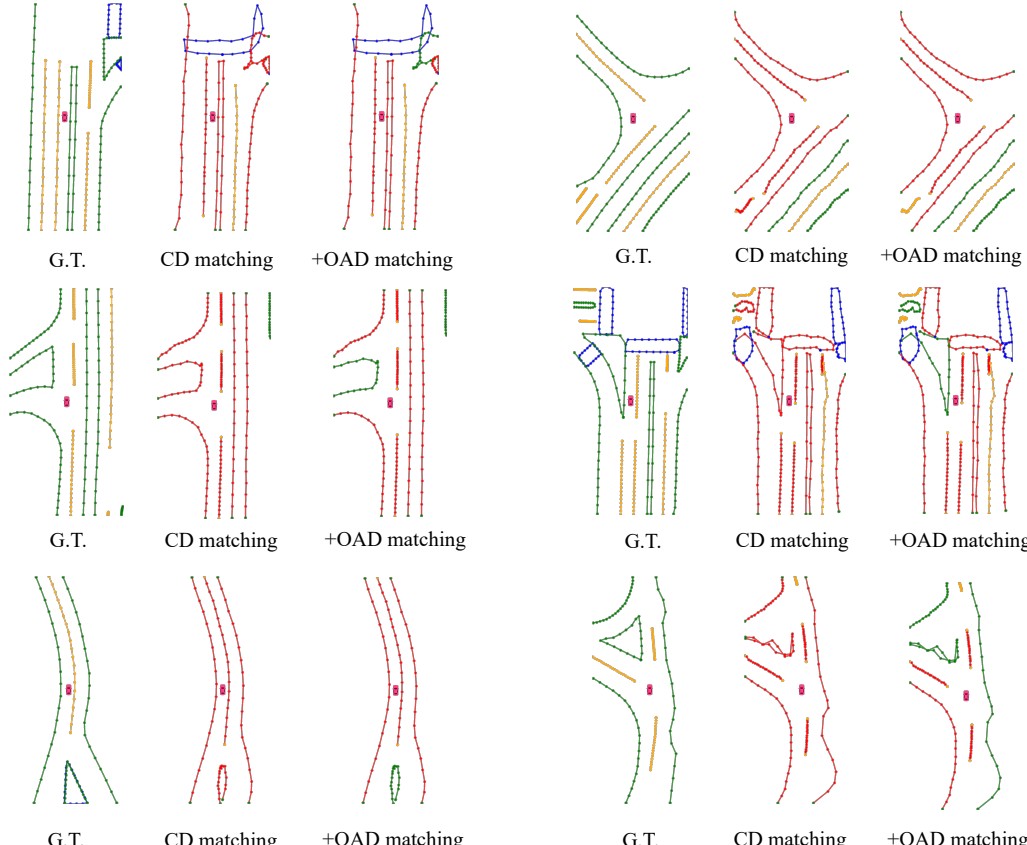

| G.T. | CD matching | +OAD matching | | G.T. | CD matching | +OAD matching |

| G.T. | CD matching | +OAD matching | | G.T. | CD matching | +OAD matching |

| G.T. | CD matching | +OAD matching | | G.T. | CD matching | +OAD matching |