# OpenReview forum: "PolyDiffuse: Polygonal Shape Reconstruction via Guided Set Diffusion Models"
_NeurIPS.cc/2023/Conference — NeurIPS 2023 poster_

### Official Review · Reviewer_uTWe · 2023-06-25

**Soundness:** 2 fair
**Presentation:** 3 good
**Contribution:** 2 fair
**Rating:** 5
**Confidence:** 4

**Summary:**

The paper presents a diffusion-based approach for reconstructing polygonal shapes from floorplan images and auto-drive sensor images. The idea is to refine an initial rough reconstruction through iterative denoising steps which revert a forward process that diffuses regular reconstructions into random noises through per-element noise injection (though the notions of elements are not clearly given for different tasks).  The challenge with learning such a denoising diffusion model, as identified by the paper, is the robustness to permutations of orders of elements that are introduced by linearization (though it's not convincing why a set-based transformer model would introduce such ordering problem inevitably). The paper proposes to address this challenge by learning a guided forward diffusion process, such that the diffusion paths for different permutations are encouraged to be separate. In test time, the rough initial reconstruction is first converted to corresponding target noise distributions by the learned guided diffusion networks, and then reverted back to an accurate reconstruction through iterative denoising.

The approach is tested on two polygonal shape reconstruction tasks, including floorplans and autonomous driving maps, and shows improvements over the baselines that produce initial reconstructions. Ablation studies are conducted on diffusion guidance and choices of several hyper parameters.

**Strengths:**

The paper proposes to improve polygonal reconstruction works that produce results in a single network pass by iterative denoising of the reconstruction. It is thus expected that the results can have better quality than the initial estimates, which is confirmed by tests on two tasks.

The paper formulates the diffusion model for structured elements like polygons and polylines, and identifies the problem of ordering permutation in representing the structures, which may cause severe ambiguity for a network to learn the reverse process.

The paper designs a learned guided forward diffusion process to distinguish the different permutations causing representational ambiguity, and derives the updated diffusion models biased by the learned diffusions. Such adaptive and learned diffusion can be inspiring to other situations where hand-crafting a diffusion process can be problematic.



**Weaknesses:**

The computational cost of the iterative process can be better illuminated, to help readers appreciate the comparative advantage obtained over the initial estimates produced by other methods.

The paper identifies another challenge with diffusion-based approach for reconstruction: a reconstruction task has a single solution, so the initial noise needs to be chosen carefully. I do not fully understand this statement, and cannot find any validation of this challenge in the experiments and discussions. On the other hand, the proposed guided diffusion process does not produce a single noise for an input polygonal shape either, but samples from the predicted Gaussian distribution.

The guided diffusion process is trained through contrastive learning on the initial reconstructions of training data by a different method. It is not clear how well the guided diffusion generalizes for data samples outside the training set. In contrast, a fixed diffusion procedure as adopted by DDPM and others does not have generalization issue, as the final noise distributions are known a prior. Indeed, the paper acknowledges that when the initial reconstructions come from hand annotations that are largely different from training reconstructions for HD Maps, the guided diffusion and denoiser do not work robustly.

It might be difficult to regard this work as the first one using diffusion models for reconstruction. In fact, diffusion models have been widely used for conditioned generation that resemble reconstruction, e.g. ControlNet. Also, people have explored using diffusion models for segmentation [1], which is a kind of reconstruction.

[1] SegDiff: Image Segmentation with Diffusion Probabilistic Models.

**Questions:**

1. Computational costs of the iterative denoisers should be reported, and compared with the base cost of running the initial reconstructions.

2. Please provide clarifications and experimental evidence that perturbations of the optimal noise for a particular reconstruction can cause failure of denoise-based reconstruction.

3. In the footnote of page 3, it's stated that even a permutation-invariant model like Transformer has to pick a permutation as the representation of a data sample. This statement seems self-contradictory and needs clarification. For example, in DETR like works (including RoomFormer and MapTR) there is no ordering of the elements, and they rely on matching to compute the loss functions between two sets. Why cannot this approach to extended to the diffusion setting?

4. When defining the permutation contrastive loss (12), why not compare the x_t and x'_t, but compare x_0 and x_t instead? Is it essential to constrain the similarity between x_0 the uncorrupted data and x_t the noised version? This is not common for standard diffusion models.

5. Sec.4, feature representation, it's not clear how polylines/polygons of different vertex numbers are encoded and represented. This is an important detail as its encoding directly impacts how likelihood-based refinement (Fig4) can be done.

6. The line matching metrics augmented with angle threshold is not defined clearly. There is no formal description of its computation in either text or supplemental.

7. Define the standard DM model of Sec.5.3 clearly. Otherwise it's hard to justify any proposed component.

8. What is training schedule? It looks like training epochs but why would one bother with the training iterations so much.

9. Since it is acknowledged that the guided diffusion network may not generalize to hand-annotated input, I hope the authors can thoroughly discuss the limitations of the learned diffusion network, including how to decide if it's not general enough for a particular test dataset or input pattern, and what are the practical ways to enhance its generalization scope.

**Limitations:**

Yes.

---

> ### Author Rebuttal · Authors · 2023-08-05
>
> We thank you for the valuable questions and thoughtful feedback. We answer your questions and comments as follows.
>
> (A clarification: "elements" are the polygons/polylines in our tasks)
>
> ---
> **W1&Q1. Computational costs of the iterative denoisers.**
>
> Due to the space limit, we refer to *Q3 of Reviewer t57P*  and *GR1 of the global response* for discussions and tables.
>
> ---
> **W2&Q2. Clarifications and experimental evidence that perturbations of the optimal noise for a particular reconstruction can cause failure of denoise-based reconstruction.**
>
> Table 3 of the main paper has shown quantitatively that GS-DM is better than a standard DM model that uses initial noise drawn from standard Gaussian (i.e., non-optimal noise distribution compared to the per-element Gaussian distribution estimated by our guidance network). We also included an illustrative toy experiment in *GR3 of the global response*, showing standard DM model easily fails with bad initial noises.
>
> ---
> **Q3. The footnote on page-3, and why cannot DETR's approach be extended to the diffusion setting?**
>
> We thank for a great question. Here is the clarification, which we will also be added to the paper: The use of a matching loss (e.g., the Hungarian matching loss in DETR) could make DMs permutation-invariant indeed. However, we cannot pick an arbitrary loss in the formulation. The L2 denoising loss of DMs is strictly derived from either the variational lower bound (DDPM) or denoising score matching (NCSN) and is order-aware by definition (i.e., based on one specific serialization of the data). Specifically, in DDPM's derivations, the loss consists of a sequence of KL divergences between the forward and the reverse Gaussians, which cannot be easily extended to a permutation-invariant version -- We are unaware of any DM formulation that is inherently permutation-invariant.
>
> ---
> **Q4. Eq.12: why not compare the x_t and x'_t, but compare x_0 and x_t instead?**
>
> The x_t and x'_t are permutation variants of the same noisy sample. They should not interfere (i.e., get close) with each other, which would make denoising ambiguous. We provide more explanations here and will clarify in the paper.
>
> The core motivation of GS-DM is "The guidance network learns to guide the noise injection in a per-element manner, such that a sample x_0 remains separated from its permutation variants throughout the diffusion process". The second triplet loss term in Eq.12 encourages x_t to be closer to x_0 than any other permutation-equivalent variants x'_t. x_0 is the *anchor* using the terminology of triplet loss, and x_t and x'_t are the *positive* and *negative*, respectively. In triplet loss, the positive (x_t) and negative (x'_t) are not directly compared to each other, but compared against the anchor (x_0) respectively. Please see Sec.3.3 of the supp for full details of the permutation loss.
>
> ---
> **Q5. Sec.4, feature representation.**
>
> We thank for the question and will clarify in the paper. A data sample is represented by a $N\times M\times 2$ tensor. $N$ is the number of polygonal instances and $M=\max_{i=1}^N N_i$ is the maximum number of vertices in an instance. The last dimension is each vertex's (x,y) coordinates. Instances with less than $M$ vertices are padded, and a mask is generated to handle the padding in the model. The order of instances is based on an arbitrary permutation. In the model, the vertex coordinates are transformed into sinusoidal positional encodings and augmented with two additional position encodings for the vertex index and instance index. All these positional encodings are concatenated and processed by an MLP to produce the feature with shape $(N*M)\times D$, where $D$ is the hidden dimension of the denoising network (we use the same hyperparameter as the base method).
>
> ---
> **Q6. Clear descriptions of the line matching metrics augmented with angle threshold.**
>
> The full details of the angle-aware matching criterion are in *Sec.4.2 of the supplementary document*. The original average precision(AP) metric considers a predicted instance as true positive once the Chamfer-distance criterion is met. Our augmented AP considers a predicted instance as true positive only when both the Chamfer-distance and angle-distance criteria are met. We will add more explanations.
>
> ---
> **Q7. Define the standard DM model of Sec.5.3 clearly.**
>
> We thank for the question. The following is the definition, and we will clarify in the paper. The standard DM model is the vanilla DDPM with images as the condition. There is *no guidance network* and *no proposal generator*. The reverse process starts with a noise sampled from the standard Gaussian distribution and gradually denoises it into the final reconstruction using the same sampler as GS-DM. It also uses the same denoising network architecture as GS-DM. This model helps to ablate the key designs of our GS-DM.
>
> ---
> **Q8. What is the training schedule, and why bother with it so much.**
>
> "Training schedule" means the training iterations and the corresponding learning rate decay strategy. We mentioned this because diffusion models need more iterations to converge well than the one-shot counterparts (i.e., RoomFormer and MapTR). To make our comparisons as fair as possible, we increase and match the training iterations of the base methods with ours.
>
> ---
> **W3&Q9. Discussions about the limitations of the learned diffusion network, including how to decide if it's not general enough for a particular test dataset or input pattern, and what are the practical ways to enhance its generalization scope.**
>
> Due to the space limit, we refer to our answer in *Q2 of Reviewer pHyL* for thorough discussions and analyses of failure modes. Please also see extended discussions with concrete examples on the HD mapping task in *GR2 of the global response*.
>
> ---
> **W4. The first one using diffusion models for reconstruction.**
>
> We thank for the comment. We will revise the claim and limit the scope to geometry reconstruction.

---

> > ### Comment · Reviewer_uTWe · 2023-08-15
> >
> > Thanks for the detailed response. I hope the authors can incorporate the updates and discussions to the final version.

---

### Official Review · Reviewer_Fcug · 2023-07-02

**Soundness:** 3 good
**Presentation:** 3 good
**Contribution:** 3 good
**Rating:** 6
**Confidence:** 4

**Summary:**

This paper proposes a Guided Set Diffusion Model for reconstruction, addressing the challenges of ambiguous denoising and selecting appropriate initial noise. By learning guidance networks, the model ensures distinct representations for samples with multiple permutations in structured geometry. During testing, the model uses the guidance networks to initialize Gaussian noise and denoises to reconstruct polygonal shapes conditioned on sensor data. The approach is evaluated on floorplans and HD maps, demonstrating significant advancements over existing methods and enabling practical applications.

**Strengths:**

The paper takes a step towards formulating reconstruction as a generation process conditioned on sensor data, providing insights into advancing Diffusion Models for shape reconstruction. The approach of guiding noise injection per element to ensure separation from permutation variants during the diffusion process effectively addresses the ambiguous denoising issue. The PolyDiffuse method is well-presented and achieves promising results for layout reconstruction. Overall, this paper makes valuable contributions and is commendable.



**Weaknesses:**

While floorplan and HD map reconstruction are important problems, it appears that the GS-DM formulation is more suitable for layout or other semantic reconstruction tasks rather than real 3D shape reconstruction. It would be beneficial if the authors could provide a discussion on this aspect and given some suggestion of addressing the limitations of PolyDiffuse in extending to 3D shape reconstruction tasks.

**Questions:**

N/A

---

> ### Author Rebuttal · Authors · 2023-08-05
>
> We thank you for the valuable comments and appreciate the overall positive feedback.
>
> The question of extending PolyDiffuse to 3D reconstruction is quite open-ended and worth a detailed discussion. We provide our thoughts in the following, and the answer is organized into two parts: 1) describe the settings of 3D shape reconstruction tasks where extensions of PolyDiffuse are possible; 2) explain the key challenges of applying PolyDiffuse to these tasks in the current stage. We will also add discussions for future works to the paper.
>
> ---
>
> **1. Task settings:** Since PolyDiffuse reconstructs the data as a set of polygonal structures, its potential extensions to 3D shape reconstruction will be reconstructing 3D CAD models or compact low-poly meshes. We briefly describe two related lines of previous works for indoor scenes and objects, respectively:
>
> (1). *Indoor CAD reconstruction:* This line of work aims to reconstruct CAD-quality models from RGB-D captures of a large indoor scene, facilitating valuable applications in the construction industry. Typical works include Structured Indoor Modeling[R1] and some of its follow-ups[R2, R3].  The entire scene usually contains many rooms, and the rooms consist of a set of planar polygonal structures for walls, floors, and ceilings.
>
> (2). *Object shape reconstruction:* Another line of work studies generative models of objects (usually from a single category) as compact low-poly meshes. As these generative models can naturally be paired with encoders (for point clouds or images) to support conditional generation, they are also able to reconstruct object shapes from sensor inputs. From our knowledge, typical works include PolyGen[R4] and BSP-Net[R5]. The compact meshes can be regarded as a set of variable-length polygons, while the polygons efficiently share the vertices.
>
> Given the similarities in the data representation, the above two lines of work are potential directions to extend PolyDiffuse.
>
> ---
> **2. Key challenges:** (1). the lack of large datasets is an immediate challenge, as paired data with sensor inputs and CAD-quality annotations is very rare, even considering synthetic ones. PolyGen[R4] already observed overfitting issues when training on pre-processed ShapeNet data. Previous works on indoor CAD reconstruction [R1-R3] rely on heuristics to produce the final compact CAD-quality models, and there are no end-to-end deep learning methods due to the lack of training data; (2). While PolyDiffuse is good at reconstructing the geometry of the polygonal shapes (i.e., the accurate coordinates of vertices), it needs a reasonable proposal generator to provide the "meta-information" -- the number of polygons and the number of vertices per polygon. While getting reliable meta-information is not hard in floorplan and HD map reconstructions, it becomes more challenging for large-scale indoor scenes or complicated objects (e.g., there might be curved surfaces).
>
> ---
>
> In summary, extending GS-DM and PolyDiffuse to more challenging 3D reconstruction tasks is an interesting yet challenging future direction. We need large datasets as well as some reliable approaches to estimate the meta-information.
>
> ---
>
> ### References
>
> [R1]. Ikehata, Satoshi, Hang Yang, and Yasutaka Furukawa. "Structured indoor modeling." In Proceedings of the IEEE international conference on computer vision, 2015.
>
> [R2]. Macher, Hélène, Tania Landes, and Pierre Grussenmeyer. "From point clouds to building information models: 3D semi-automatic reconstruction of indoors of existing buildings." Applied Sciences 7, no. 10 (2017): 1030.
>
> [R3]. Tang, Shengjun, Xiaoming Li, Xianwei Zheng, Bo Wu, Weixi Wang, and Yunjie Zhang. "BIM generation from 3D point clouds by combining 3D deep learning and improved morphological approach." Automation in Construction 141 (2022): 104422.
>
> [R4]. Nash, Charlie, Yaroslav Ganin, SM Ali Eslami, and Peter Battaglia. "Polygen: An autoregressive generative model of 3d meshes." In International conference on machine learning, pp. 7220-7229. PMLR, 2020.
>
> [R5]. Chen, Zhiqin, Andrea Tagliasacchi, and Hao Zhang. "Bsp-net: Generating compact meshes via binary space partitioning." In Proceedings of the IEEE/CVF Conference on Computer Vision and Pattern Recognition, pp. 45-54. 2020.

---

> > ### Comment · Reviewer_Fcug · 2023-08-21
> >
> > Thanks for answering my questions.

---

### Official Review · Reviewer_t57P · 2023-07-05

**Soundness:** 3 good
**Presentation:** 3 good
**Contribution:** 3 good
**Rating:** 6
**Confidence:** 4

**Summary:**

This paper proposes a novel method for reconstructing multiple polygon shapes using a conditioned diffusion model. The method first learns a score-matching-based prior diffusion model from data. This model is then used to denoise the sensor data, resulting in the reconstruction of the polygon shapes. To handle the permutation order of the polygonal elements, the method separates the target Gaussian distribution of a sample from its permutation variants.

**Strengths:**

1. A nice solution to handle the permutation order in the reconstruction of multiple geometric elements in conditioned diffusion model.
2. Comprehensive experiments to evaluate the performance of the proposed method.
3.This paper is well-written and easy to follow. Figure 2 provides a clear and concise illustration of the basic idea.


**Weaknesses:**

1. The maximum number of polygons and their vertices used in the training of the diffusion model should be clarified. Are these numbers consistent with the ones listed in lines 183-184?



**Questions:**

Overall, I like the idea of integrating a diffusion model as a prior into the reconstruction pipeline. However, I wonder if it is possible to handle the permutation order in a different way. For example, we could first train a network to predict the structure of the floorplan, such as the number of rooms and their basic bounding boxes. Then, we could denoise each shape individually. This approach might be easier to train and faster in inference.

**Limitations:**

The authors have discussed the limitations of the proposed method in Sec. 7.  I also want to know whether the denoising speed is fast enough for a real-time HD map reconstruction.

---

> ### Author Rebuttal · Authors · 2023-08-04
>
> We thank you for the constructive questions and appreciate the overall positive comments on our presentation, idea, and experiments. We answer your questions/concerns as follows.
>
> ---
>
> **Q1. The maximum number of polygons and their vertices used in the training of the diffusion model should be clarified.**
>
> Thank you for the catch. We provide the details and will also clarify them in the paper. For the floorplan reconstruction task, the maximum number of polygons and maximum vertices per polygon used during training are 20 and 40, respectively. These choices are inherited from the base method RoomFormer and are actually larger than the maximum possible numbers of the Structured3D dataset (reported in L183-184).  For the HD map reconstruction task, the maximum number of polygons/polylines and maximum vertices per instance used during training are 30 and 20, respectively. Note that we employ the same map representation as the base method MapTR -- all the map instances have 20 uniformly interpolated vertices.
>
> At test time, PolyDiffuse takes whatever the proposal generator produces and does not limit the maximum number of polygons and vertices per polygon.
>
> ---
>
> **Q2. If it is possible to handle the permutation order in a different way. For example, we could first train a network to predict the structure of the floorplan, such as the number of rooms and their basic bounding boxes. Then, we could denoise each shape individually. This approach might be easier to train and faster in inference.**
>
> This is an interesting idea -- first training a simple network to predict the "meta-information" of the structures (i.e., number of instances, bounding box of each instance, and the rough number of vertices), and then employing a diffusion model to denoise the meta-information of each instance individually (should be in-parallel denoising for acceleration). Since the shapes are denoised separately, there is no bothering with the set ambiguity issue. However, we have two concerns below, which might potentially limit the performance:
>
> (1). Some pre-processing steps might be necessary for employing a diffusion model to denoise each polygonal instance separately. For example, we might need to normalize the coordinate space and crop image features based on the bounding box of each instance, such that the denoised polygons are bounded by the boxes and do not interfere with each other. The ground-truth bounding boxes can be used at training time. But at test time, recovering from an inaccurate initial bounding box (e.g., a too-small box) could be hard.
>
> (2). The inter-instance relation is a crucial type of pattern in structured polygonal data. For example, neighboring polygons effectively share corners and edges (as in floorplan), and polylines are usually parallel, orthogonal, or connected to each other based on certain regularities (as in HD maps). The individual design makes it impossible for the denoising network to model the inter-instance interactions, which could largely impair the performance. To alleviate this issue, we might need additional clever designs to allow message passing between the instances during the in-parallel denoising. On the contrary, our PolyDiffuse can directly borrow the network architecture of state-of-the-art one-shot methods (e.g., RoomFormer, MapTR) to implement the denoising network.
>
> (3). Using bounding boxes as the intermediate representation is a bit task-specific and might not support arbitrary initial inputs (e.g., human scribbles as rough annotations). On the contrary, the guidance network in our GS-DM formulation is very general: 1) the training simply relies on a distance-based loss without specifying a typical type of input; 2) it can take different initial results at test time (i.e., either the results from existing methods or arbitrary human inputs).
>
> Overall speaking, we believe this idea is feasible and can be faster than the current formulation of PolyDiffuse w/ GS-DM. But given the above concerns, more complicated designs will be needed to get a satisfactory performance, and the method will be a bit task-specific.
>
> ---
> **Q3. Whether the denoising speed is fast enough for a real-time HD map reconstruction.**
>
> We thank you for the question and have measured the running time of PolyDiffuse based on MapTR-tiny (using ResNet-50 image backbone as in Table.2 of the main paper). We provide the details in this rebuttal and will include related information in the paper.
>
> The running time is measured with *a single Nvidia RTX A5000 GPU* on our machine. The time used by the image encoding (i.e., processing six perspective images with the ResNet and aggregating all features into the BEV space by a Transformer) is **roughly four times** the time used by the Transformer decoder to produce the final denoising outputs. Since all the denoising steps share the same BEV features, the image encoding will only run once at test time.
>
> **FPS stats:** MapTR-tiny has 14.3 FPS in our computation environment. With the same computing resources, a 5-step PolyDiffuse has 7.2 FPS, and a 10-step PolyDiffuse has 4.4 FPS. If we count both the running time of the MapTR proposal generator and PolyDiffuse, the results are 4.8 FPS for 5-step and 3.4 FPS for 10-step.
>
> We also provide an updated table in *GR1 of the global response* to better demonstrate the "running time vs. performance" relation for both tasks. Please consider taking a look there.

---

> > ### Comment · Reviewer_t57P · 2023-08-20
> > **Thx**
> >
> > Thanks for clarifying my questions. I have no further concerns.

---

### Official Review · Reviewer_pHyL · 2023-07-06

**Soundness:** 4 excellent
**Presentation:** 3 good
**Contribution:** 3 good
**Rating:** 7
**Confidence:** 3

**Summary:**

- The paper introduces PolyDiffuse, a novel structured reconstruction algorithm that incorporates Guided Set Diffusion Models (GS-DM).
- The core concept involves splitting the reconstruction pipeline into two distinct stages.
   - The forward diffusion process focuses on learning guidance networks to address denoising ambiguity effectively.
   - The guidance networks establish individual Gaussians as target distributions for each polygon and are learned prior to the denoising training phase.
   - The reverse denoising process utilizes the sensor data as a condition to reconstruct polygonal shapes.
- The authors conduct extensive experiments involving two polygonal structure reconstruction tasks.
- Through comprehensive quantitative and qualitative evaluations, the results demonstrate that GS-DM outperforms existing state-of-the-art methods in terms of performance and quality of the reconstructions.

**Strengths:**

- The paper is well-written and easy to follow.
- The derivation of the GS-DM is appropriately detailed.
- By employing a multi-stage approach and introducing guidance, the proposed method effectively resolves ambiguity in polygonal shape scenarios.
- The proposed method can evaluate reconstruction quality through likelihood evaluation.
- The proposed method performs well even when using only rough annotations as input.



**Weaknesses:**

The GS-DM may be sensitive to the specific guidance, as mentioned in Section 7 (Limitations).

**Questions:**

- It is worth exploring whether incorporating the condition $y$ as an input to the guidance network leads to performance improvements.
- What are the failure cases when applying PolyDiffuse to off-the-shelf methods? Is there further analysis regarding these cases?

**Limitations:**

The authors addressed the limitations.

---

> ### Author Rebuttal · Authors · 2023-08-04
>
> We thank you for the time and effort in providing insightful feedback and appreciate the overall positive comments. We answer the questions and concerns as follows.
>
> ---
> **Q1. It is worth exploring whether incorporating the condition as an input to the guidance network leads to performance improvements.**
>
> Yes, this is a potential strategy to improve the quality of the guidance. However, we chose not to do this in the current formulation due to three considerations:
>
> (1). The proposal generator (i.e., either an existing method like RoomFormer or MapTR, or a human annotator to provide rough annotations) already takes $\mathbf{y}$ as the input to produce the initial proposal $\hat{\mathbf{x}}_0$, so having both $\hat{\mathbf{x}}_0$ and $\mathbf{y}$ as inputs to the guidance network might not be necessary. Furthermore, the outputs of the guidance network are combined with Gaussian noise to produce noisy data during denoising training and are adjusted by the denoising network to get the final reconstruction during sampling, so they do not have to be precise.
>
> (2). Incorporating the sensor input $\mathbf{y}$ could bring significant computation costs since it requires image encoding and cross-attention layers (for extracting image features). Our current guidance network is simply a lightweight Transformer with two self-attention layers;
>
> (3). The guidance training loss is designed to preserve the permutation of the input data in the forward process, and there is no regression or classification-style supervision as in single-shot methods (e.g., RoomFormer and MapTR). With this loss design, the guidance network might not effectively use the information of $\mathbf{y}$.
>
> ---
> **Q2. What are the failure cases when applying PolyDiffuse to off-the-shelf methods? Is there further analysis regarding these cases?**
>
> Three types of failure modes are observed in our experiments, and they are mentioned and roughly explained in the main paper (Sections 5.1, 5.2, and 7). We will provide more detailed discussions in the following and will also add them to the paper. Please also see extended discussions in *GR2 of the global response*, where we provide a figure to show concrete examples of the failure cases on the HD mapping task.
>
> (1). **Wrong number of vertices.** This type of failure happens when the proposal generator predicts wrong vertex numbers for the polygons/polylines, which has been shown in Fig.4 of the main paper. We have provided a feasible solution to address this failure mode by leveraging the likelihood evaluation property of our DM-based method, where we can locally search different vertex numbers for the polygons and get the one with the highest likelihood as the final reconstruction result.
>
> (2). **Wrong number of polygonal instances (i.e., elements of the set).** As mentioned in the Limitations in Sec.7, this type of failure case happens when the proposal generator misses entire polygons/polylines, or predicts redundant elements (e.g., the ground truth is a large polygon, but the proposal generator predicts two separate small polygons).
>
> A potential solution for the former case is still based on the likelihood evaluation property: i) Run the proposal generator with a lower confidence threshold to get an initial result with more instances, which has a better recall but lower precision; ii) Extract subsets of the initial result as different initial proposals for PolyDiffuse, run PolyDiffuse and evaluate the likelihood of the final reconstruction; iii) Pick the one with the highest likelihood as the final result. This strategy could potentially recover missing instances but needs much more running time.
>
> The latter case is more challenging, and PolyDiffuse cannot rectify these redundant elements in the initial proposals. We hope there can be future works to handle the number of instances in an elegant way.
>
> (3). **Inaccurate shape/location caused by limited generalization ability of the networks to unseen styles of initial proposals.** This type of failure case is also mentioned in the Limitations subsection of Sec.7. Concretely, in our current training settings, the guidance network is trained only with "ground-truth proposals" during the guidance training stage (Algorithm1 in the paper), and it also only takes the "ground-truth proposals" to produce the guidance for training the denoising network during the denoising training stage (Algorithm2 in the paper). As a result, the networks might not perfectly generalize to other styles of the initial proposals, like the circle-shaped rough annotations used in our experiments, which can lead to *inaccurate location or shape* of the final reconstruction.
>
> One potential solution to alleviate this generalization issue is to train the guidance network and denoising network with different types of initial proposals at training time by augmenting the ground-truth data. Possible augmentation strategies include: adding noise to the vertex coordinates, or converting the G.T. elements into some canonical shapes (e.g., circles as in our experiments) to mimic the style of rough annotations, etc.
>
> Another potential solution is to train separate guidance and denoising networks for each type of initial proposal and applies the corresponding models at test time with respect to the exact type of initial reconstruction. But this solution will induce more computational cost for training.
>
> (Note: we use "initial proposal" and "initial reconstruction" interchangeably in the above descriptions)

---

> > ### Comment · Reviewer_pHyL · 2023-08-21
> >
> > Thanks for addressing my concern with a detailed response.

---

### Official Review · Reviewer_HnVz · 2023-07-07

**Soundness:** 4 excellent
**Presentation:** 4 excellent
**Contribution:** 4 excellent
**Rating:** 7
**Confidence:** 3

**Summary:**

The manuscript introduces a adaptation of the DDPM paradigm to enable denoising sets (instead of single data points like images). The method called GS-DM does this by introducing by adding noise per set element via learned guidance networks.
This approach requires the addition of a proposal generator that initializes the number of sets and their parameters for the denoising process.
The manuscript applies this new set-based diffusion model to two tasks: generating floorplan polygons from floorplan images and maps from autonomous car images. The proposed method performs well in comparison to related work.



**Strengths:**


The authors identify a core limitation in DDPMs when it comes to set-based data and propose a sound way that empirically works well to solve it. The paper is well written, figures are high quality and support the writing well.
The evaluation is thorough and clearly demonstrates that (1) the set-based diffusion is superior to the default DDPM (table 3) and (2) that the denoising approach helps improve state of the art results from MapTR and RoomFormer when applied on top of their final outputs. It is also very interesting to see how simple the inputs for denoising can be (as simple as the polygon centers) and still perform at almost state of the art levels.

Since the underlying concept of set diffusion models is very general and could apply in various other domains, this work is significant to a larger audience than just the 3d reconstruction or perception community.


**Weaknesses:**


While the paper is well written it could use some more clarity around how the guidance network is learned(line 122ff) since this is one of the key novelties. Just from reading the text I would not be able to reproduce the method at this point.
- How is the neares negative permutation found? It seems like this could be an intractable problem very quickly?
- is the triplet loss computed based simply on the polygon corners (for example?)

The key limitation of the current approach is the fact that the number of elements in the set has to be initialized correctly by the proposal generator during inference. This is acknowledged by the authors and a reasonable limitation that can be addressed in follow on work imo.



**Questions:**

- In Fig3: is it mu/sigma _phi or _psi?


**Limitations:**

The limitations are adequately addressed by the authors at the end of the paper.

---

> ### Author Rebuttal · Authors · 2023-08-03
>
> We thank you for the valuable comments and appreciate the overall positive feedback on the writing, the experiments, and the potential extension of our approach to broader domains. We will address your questions/comments as follows.
>
> ---
>
> **Weaknesses: Clarifications of the guidance network learning.**
>
> We are sorry for the confusion. Details are presented in the supplementary (in particular, the permutation loss Eq.12 in *Sec.3.3 of the supplementary document*). We briefly answer your two sub-questions below and will add these clarifications to the main paper.
>
> (1). **nearest negative permutation in the permutation loss**: Yes, finding the nearest negative permutation immediately becomes intractable as the number of elements ($N$) grows. As described in Sec.3.3 of the supplementary document, our solution is to approximate the permutation loss (Eq.12 of the main paper) with $N$ element-level triplet losses (Eq.9-12 of the supp), which reduces the computational cost from $O(N!)$ to $O(N^2M)$, where $M$ is the maximum number of vertices of an element (i.e., a polygon or polyline) in the data sample $\mathbf{x}_0$. Empirically, our experimental results show that this approximation works well and helps learn reasonable guidance under feasible computations.
>
> (2). **Triplet loss computation:** The triplet loss is computed based on the L1 distance between the corners' coordinates of the two polygons or polylines (the $D(i, j)$ in Sec3.3 of the supp). We will clarify this detail.
>
> ---
>
> **Questions: Symbols in Fig.3.**
>
> It should be phi for both bar_mu and bar_sigma in Fig.3. We thank you for the catch and will fix the mistake.

---

> > ### Comment · Reviewer_HnVz · 2023-08-14
> > **response**
> >
> > Thanks for clarifying my questions.

---

### Author Rebuttal · Authors · 2023-08-06

We thank all reviewers for your time and efforts in providing valuable comments and constructive feedback. We are glad that all the initial reviews are on the positive side (two accept, two weak accept, and one borderline accept).

---
We use this global response as **a complement to the individual responses**, which will answer reviewers' questions that require additional figures or tables. The content of this global response is organized as follows, and we will also incorporate them into the paper.

**GR1:** Tables and discussions for the computational costs of PolyDiffuse on the two tasks. **(for Reviewer-t57P and Reviewer-uTWe)**

**GR2:** A figure of PolyDiffuse's example failure cases on the HD mapping task, as well as corresponding discussions. *This is an extension of our response to Q2 of Reviewer-pHyL*. **(for Reviewer-pHyL and Reviewer-uTWe)**

**GR3:** An illustrative toy experiment to show how standard DM fails, with a figure and discussions. **(for Reviewer-uTWe)**

---
**GR1. (Reviewer-t57P, Reviewer-uTWe) The computational cost of the denoising process, and the comparison to the methods for generating initial proposals.**

Table 1 and Table 2 in the rebuttal PDF present the speed-performance tradeoff of PolyDiffuse on the HD map and floorplan reconstruction tasks, respectively. Note that during the denoising process, the image encoding parts of the denoising network only run once, while the transformer decoder part runs for multiple rounds. In our computation environment, *the time of image encoding vs. the time of transformer decoder* is 4:1 for HD map reconstruction (w/ MapTR's network architecture) and 2:1 for floorplan reconstruction (w/ RoomFormer's network architecture).

Speed is not a crucial factor for floorplan reconstruction, so we can just use more denoising steps in real applications for better reconstruction quality. For HD map reconstruction, since FPS is an important consideration for online applications, we might have to pick the number of denoising steps more carefully. In general, our method provides a reasonable speed-performance tradeoff. Furthermore, since PolyDiffuse is not restricted to a fixed task-specific model for the proposal generator, our performance/efficiency can keep improving when better task-specific base methods appear in the future.

---
**GR2. (Reviewer-pHyL, Reviewer-uTWe) Extended discussions/analyses on the failure modes of PolyDiffuse.**

*(This response is an extension of our response to Q2 of Reviewer-pHyL, please read that response first)*

We have discussed three types of failure modes in *Q2 of Reviewer-pHyL*. The first type (i.e., wrong number of vertices) has been covered in *Figure 4 of the main paper*. We now use Figure 1 of the rebuttal PDF to show the other two types of failure cases with the HD mapping task and provide some discussions below.

**Wrong number of instances.** The examples (1) (2) (3) in Figure 1 of the rebuttal PDF are typical failure modes of having a wrong number of instances. When MapTR serves as the proposal generator, the mistakes of missing instances or predicting duplicate/redundant instances are hard to be recovered by PolyDiffuse. Our *response to Q2 of Reviewer-pHyL* only provides an alleviating strategy based on search and likelihood evaluation, and more elegant approaches are needed to better handle this challenge. Also, if the proposal generator predicts wrong semantic labels, PolyDiffuse is not able to recover.

**Inaccurate shape/location caused by limited generalization ability of the networks to unseen styles of initial proposals.** When using rough human annotations as the initial proposals in the paper, we assume that the correct number of polygonal instances and semantics labels are given, and PolyDiffuse is responsible for generating accurate coordinates for all the vertices. However, as demonstrated by (4) (5) (6) of Figure 1, although the results are visually reasonable, many predictions are not considered as true positive by the matching criteria.

As analyzed in our *response to Q2 of Reviewer-pHyL*, the networks don't perfectly generalize to the circle-shaped initial proposals, leading to slight shifts in instance location or errors in shape. Note that the matching criteria of the HD mapping task are very strict (autonomous driving must guarantee safety). This inaccurate shape/location issue also appears when using MapTR initial proposals due to some inaccurate initial predictions, but the overall precision of using MapTR proposals is higher than using rough annotations, as shown in Table 2 of the main paper.

Another cause of inaccurate location/shape is the noisy image inputs caused by occlusions or bad weather conditions, but this is a common challenge for all HD mapping methods (e.g., MapTR, VectorMapNet, etc.) and is out of the scope of this paper.

---

**GR3. (Reviewer-uTWe)  Clarifications and experimental evidence that perturbations of the optimal noise for a particular reconstruction can cause failure of denoise-based reconstruction.**

In Figure 2 of the rebuttal PDF, we provide a toy experiment to demonstrate how standard DM models easily fail and why a good initial noise is important. Note that we have clarified the definition of standard DM in the *response to Q7 of Reviewer-uTWe*. In this example, the data contains a single toy example with 6 rectangular shapes, so there are $6! = 720$ permutation-equivalent representations. After sufficient training, we draw four samples using the image-conditioned denoising process. The DDIM sampler is used with 10 sampling steps, so the randomness only comes from the initial Gaussian noise. As the figure shows, only Sample 3 gets the correct reconstruction result. With the challenges of set ambiguity, a standard conditional DM even has trouble overfitting a single data sample, and easily gets wrong outputs when the initial noise is inappropriate.

---

### Decision · Program_Chairs · 2023-09-21

**Decision:**

Accept (poster)

**Comment:**

The paper has achieved a consensus for acceptance. Initially, reviewers were positive about both the technical novelty and the results presented. The rebuttal resolved most minor concerns and issues related to clarity. The AC agrees that the technical contribution of this paper, particularly the diffusion model on polygonal shapes with special care on handling permutation order, is clearly articulated. The results effectively support the paper's claims. Therefore, acceptance should be granted. Congratulations!